

# Revisiting snow settlement with microstructural knowledge

Louis Védrine[1] and Pascal Hagenmuller[1]

[1]Météo-France, CNRS, Univ. Grenoble Alpes, Univ. Toulouse, CNRM, Centre d'Études de la Neige, 38000 Grenoble, France.

**Correspondence:** Louis Védrine (louis.vedrine@meteo.fr)

**Abstract.** Snow settlement under gravity is primarily driven by the slow creep of its ice matrix, which exhibits a viscoplastic behaviour. Knowledge of the viscoplastic properties of snow is thus crucial for understanding and predicting the snowpack seasonal and perennial evolution. However, different approaches have yielded disparate constitutive viscoplastic laws. Field experiments and snowpack models typically described snow settlement with a linear model and an apparent compaction viscos-

ity. Dedicated laboratory experiments exhibited non-linear relationships between stress and strain rate, with a stress exponent ranging from 1.8 to 4. Microstructure-based simulations showed that the viscoplastic behaviour likely results from the interaction of glides on various intra-crystalline slip systems within ice crystals, yielding an exponent between 2 and 3. The paper aims to reconcile these approaches. To do so, we conducted microstructure-based simulations on 37 three-dimensional snow images and established that the viscoplastic behaviour follows a power-law relation, with a stress exponent almost constant around

2.15, and a reference stress that depends mostly on the solid fraction. Analysing a dataset from previous viscoplastic tests (178 points) revealed that applying the stress exponent from the simulations significantly reduces variability in the reference stress between independent studies and led to a simplified constitutive relation. Lastly, we showed that the linear settlement laws of snowpack models, such as Crocus, align with the proposed constitutive relation under natural loading conditions typically encountered on alpine sites, due to correlations between stress and density. However, considerable differences emerge under

"non-standard" scenarios, such as elevated loads on light snow or reduced loads on dense snow, where our model demonstrates superior qualitative performance.

KEYWORDS: Snow, Settlement, Viscoplasticity, Creep

## 1   Introduction

Under its weight, snow naturally settles from an initial density of approximately 100 kg m$^{-3}$ just after deposition on the ground

to about 500 kg m$^{-3}$ at the end of the winter season, and up to the ice density for perennial snow on glaciers and ice sheets. Snow settlement is a key process of the snowpack evolution (Brun et al., 1992; Lehning et al., 2002; Simson et al., 2021). Predicting snow density is thus essential for many applications such as avalanche forecasting (Morin et al., 2020), hydrology (Barnett et al., 2008), surface mass balance in polar regions (Morris and Wingham, 2014), or paleoclimatology (Barnola et al., 1987; Goussery et al., 2004).

Dry snow is a two-phase material made of sintered ice crystals and air (Roscoat et al., 2011). Its settlement is primarily driven by the slow viscoplastic deformation of its ice skeleton, leading to a progressive reduction in pore volume (e.g., Yosida,





1955). Indeed, snow is an open-pore material and the viscous flow of air during deformation can be neglected at low strain rates (e.g., Bergen, 1968; Colbeck, 1989). Snow settlement thus depends solely on the spatial arrangement of the ice crystals and the deformation mechanisms they undergo. Two main mechanisms are generally considered for ice in snow: intra-crystalline

deformation, also known as dislocation glide (Duval et al., 1983; Louchet, 2004; Weikusat et al., 2017), and inter-crystalline deformation, also known as grain boundary sliding (Alley, 1987; Lundin et al., 2017). At macroscale, the viscoplastic behaviour of snow, or more widely of porous polycrystals, is generally modelled by a power-law (Glen's law) relating stress $\sigma$ to strain rate $\dot{\epsilon}$ (Glen, 1955):

$$\dot{\epsilon} = \dot{\epsilon}_0 \left( \frac{\sigma}{\sigma_0} \right)^n \tag{1}$$

with $\dot{\epsilon}_0 = 1 \ \mathrm{s}^{-1}$, $n$ the stress exponent and $\sigma_0$ the reference stress. This model is extremely sensitive to $n$, which quantifies the dependence of the viscous deformation to the applied stress.

To interpret experimental compression tests on snow, the concept of compaction viscosity has often been employed (Bader, 1954, 1962; Mellor, 1974; Navarre, 1975; Gray and Morland, 1995; Kominami et al., 1998; Sturm and Holmgren, 1998; Scapozza and Bartelt, 2003a). The compaction viscosity $\eta$ linearly relates the two directly-measurable macroscopic quantities,

i.e., stress $\sigma$ and strain rate $\dot{\epsilon}$:

$$\dot{\epsilon} = \frac{\sigma}{\eta}. \tag{2}$$

This model implicitly assumes that $n = 1$ in Eq. 1 and corresponds to a Newtonian compressible viscous fluid. Several authors have encapsulated into $\eta$ various processes presumably contributing to snow creep, such as gravitational settling, grain rearrangement, water infiltration, and refreezing (Shinojima, 1967; Kojima, 1967; Bergen, 1968; Keeler, 1969). Rheological

models, for instance, the Maxwell, Kelvin, or Burger models (a combination of linear springs and dampers in parallel or series), have been the most popular to describe snow behaviour at low strain rates (Bader, 1962; Mellor, 1974; McClung and Larsen, 1989; Bartelt and Christen, 1999; Chandel et al., 2007; Gorynina et al., 2024; Bahaloo et al., 2024; Huo et al., 2024) . Nevertheless, these models apply to a wide range of conditions only if $\eta$ does not depend on $\sigma$ and $\dot{\epsilon}$, which remains questionable. Consequently, non-linear extensions of these viscoelastic models have been proposed (Desrues et al., 1980; Stoffel

and Bartelt, 2003; O'Connor and Haehnel, 2020). Besides, in line with this historical data and models, the settlement laws currently implemented in snowpack models, such as Crocus (Brun et al., 1992; Vionnet et al., 2012) and Snowpack (Lehning et al., 2002; Simson et al., 2021), are also linear. In these models, the compaction viscosity is parametrized as a function of temperature and snow microstructure (primarily its density) and has been calibrated and evaluated under specific conditions (Ganju et al., 1999). Again, it is questionable whether the implemented models apply to conditions that deviate from the cali-

bration conditions, typically a natural snowpack evolving under alpine conditions. For instance, Royer et al. (2021); Woolley et al. (2024) have tuned the standard settlement laws to represent the evolution of snowpack density under arctic conditions, but they lack a physical basis and a wealth of calibration data.

Different laboratory experiments have focused on characterizing $n$ on snow. Narita (1980, 1984) reported a stress exponent of $n = 4 \pm 1$ which appeared independent of temperature. Scapozza and Bartelt (2003a, b) conducted strain rate-controlled



triaxial experiments on alpine snow and identified two distinct regimes: regime I with an exponent of approximately $n \approx 1.8$ for low-density snow, and regime II with $n \approx 3.5$ for higher density snow, with a transition density depending on temperature. In line with this finding, Schleef et al. (2014a) reported $n \approx 2$ for compression experiments on low-density samples. Delmas (2013) observed a sensitivity of $n$ to snow types: $n = 4.1$ for faceted grains compared to $n = 3.2$ for rounded grains, at $-15°$. Recently, Sundu et al. (2024b) performed cyclic loadings on snow samples and derived $n$ from the loading and relaxation phases. They found a transition of $n$ as a function of the geometrical grain size: from $n \approx 1.9$ for small grains to $n \approx 4.4$ for large grains. They attributed this transition to a transition from grain boundary sliding to dislocation creep at a critical grain size. In general, the reported scatter of $n$ between 1.8 and 4 was often attributed to the driving mechanisms at the micro-scale, whose activation may depend on density, optical diameter, or snow type. However, experimental evidence for the reasons behind this sensitivity is still lacking (Ignat and Frost, 1987; Meyssonnier et al., 2009). Moreover, repeated experiments under perfectly controlled conditions (e.g., changing the applied stress without altering any parameter that affects the reference stress $\sigma_0$) are seldom conducted and are challenging on snow.

An approach to understanding the mechanisms of snow viscoplasticity and predicting the overall behaviour relies on numerical experiments. The key idea is to model snow based on its porous and crystalline microstructure and to incorporate the local deformation mechanisms. The numerical solver then simulates the macroscopic behaviour of the sample. Thanks to higher computing power and easier access to microstructural data, various studies have followed this approach in the recent years for snow viscoplasticity, with different assumptions regarding the geometry of the ice skeleton and the driving mechanisms (Johnson and Hopkins, 2005; Theile et al., 2011; Wautier et al., 2017; Védrine et al., 2024, 2025). Johnson and Hopkins (2005) modelled snow creep with a snow microstructure described with a set of cylinders with hemispherical ends and inter-granular viscoplastic contact law. Theile et al. (2011); Wautier et al. (2017) employed a more accurate representation of the microstructure based on tomographic data, and considered only intracrystalline deformation. These studies simulated an exponent $n = 3$ and achieved reasonable agreement with experimental data. However, the validation data were either too far from the numerical conditions to perform a 1:1 evaluation (Wautier et al., 2017) or too scarce to validate the model assumptions definitely (Theile et al., 2011). In addition, a number of their modelling assumptions are now questionable a priori. Theile et al. (2011) modelled the snow microstructure as a collection of interconnected finite element beams, which inadequately captured the deformation obstacles present at the crystal boundaries. Conversely, Wautier et al. (2017) treated the ice in snow as a homogeneous isotropic polycrystalline material, neglecting the fact that ice in snow is a collection of ice crystals surrounded by air. Indeed, Védrine et al. (2024) recently demonstrated that ice in snow cannot be modelled as an isotropic material, and that the crystalline nature of ice must be explicitly considered. In general, Védrine et al. (2025) showed that, for porous polycrystals like snow, the stress exponent is sensitive to density and intracrystalline surface area. They showed that the homogenized stress exponent varies between the exponent of the weakest (basal) slip system ($n = 2$, (Duval et al., 1983; Chevy, 2008)) and the stress exponent of dense polycrystalline ice ($n \sim 3$), depending on the frustration of the crystals.

In summary, three different approaches have been used to understand and model the viscoplastic behaviour of snow, which is essential to predict its settlement:





– Field and laboratory experiments focusing on conditions typically encountered by natural snowpacks in alpine regions
have described snow settlement by a linear model ($n = 1$) and an apparent compaction viscosity $\eta$ (Eq. 2). This law is
implemented in detailed snowpack models.

– Laboratory experiments focusing on the generic material properties of snow reported a non-linear relationship between
stress and strain rate (Eq. 1) with a stress exponent $n$ between 1.8 and 4.

– Microstructure-based simulations suggested that the macroscopic viscoplastic behaviour results from the complex inter-
play between glides on different ice intra-crystalline slip systems, which yields an exponent $n$ between 2 and 3.

Our paper aims to show that, despite these apparently contradictory results, these three approaches can be reconciled. Our
starting point is the simulations of the effective viscoplastic behaviour of numerous snow microstructures with the state-of-the-
art 3D model by Védrine et al. (2025). Then, we re-interpret previously published experimental data by searching for the model
with the minimal scatter. We show that the model ends up very close to the one derived from microstructure-based simulations.
Eventually, we analyse the difference between the latter model and the settlement law implemented in Crocus and show that
they coincide under the conditions typically encountered by natural snowpacks in the Alps.

## 2 Material and Methods

### 2.1 Microstructure-based simulations

We conducted numerical full-field simulations on three-dimensional porous microstructures, using a crystal plasticity model
to describe the local deformation mechanisms of ice. These simulations are then homogenized to derive the stress exponent $n$
and reference stress $\sigma_0$ (Eq. 1) of a given snow microstructure.

#### 2.1.1 Input snow microstructures

The input of our microstructure-based viscoplastic model is a 3D image of the snow microstructure partitioned into individual
crystals. We used 37 different X-ray tomographic images acquired from previously published studies (Hagenmuller et al.,
2016; Peinke et al., 2020; Hagenmuller and Carmagnola, 2020; Fourteau et al., 2022; Bernard et al., 2023). These binary ice-
air images encompass the majority of snow types encountered in seasonal snowpacks and span a wide range of density and
specific surface area (Fig. 1). X-ray attenuation images do not distinguish between ice crystals. We thus numerically partitioned
the ice skeleton into individual grains based on geometrical criteria, following the algorithm of Hagenmuller et al. (2014b);
Peinke et al. (2020). We assumed that the detected grains correspond to the ice crystals, which is a reasonable assumption
for most types of snow according to Arnaud et al. (1998). Then, each crystal was associated with a crystalline orientation
(c-axis) randomly sampled in an isotropic distribution. Note that Védrine et al. (2024) showed that the exact random draw
does not significantly affect the overall mechanical behaviour. To reduce computation time, simulations were not performed on
the entire scanned volume at the nominal image resolution. The resolution was downscaled while ensuring that the equivalent





optical radius, $3/(\mathrm{SSA}\rho_{\mathrm{ice}})$, remains between 6 and 12 voxels. A cubic sub-volume of $250 \times 250 \times 250$ voxels was extracted
from the image. This volume contains at least 20 crystals along each edge, which ensures the sample is representative (Védrine
et al., 2024). Some input 3D images are shown as 2D slices in Figure 2. All details on these images are given in supplementary
material and associated publications.

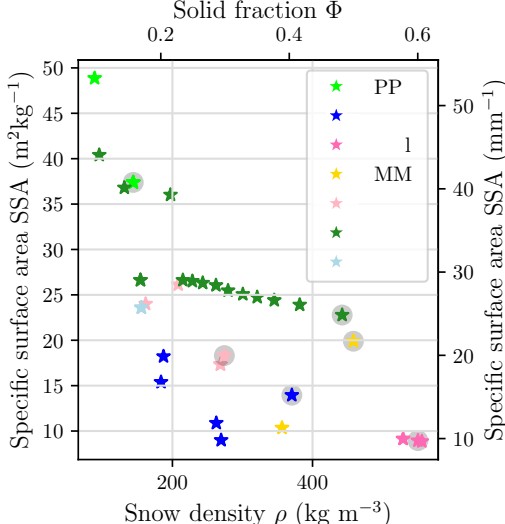

**Figure 1.** Characteristics of the 3D microstructures used in this study. The snow types are described and coloured according to the International Classification for Seasonal Snow on the Ground (Fierz et al., 2009): Precipitation Particles (PP), Decomposing and Fragmented Precipitation Particles (DF), Rounded Grains (RG), Large Rounded Grains (RGlr) Faceted Crystals (FC), Depth Hoar (DH), Machine Made (MM).The microstructures enclosed by a black circle correspond to those shown in Figure 2.

### 2.1.2   Elasto-viscoplastic simulations

We performed elasto-viscoplastic simulations using the model developed by Védrine et al. (2024, 2025). We recall here its
main assumptions. Details can be found in the appendix A and associated references.

     Air was modelled as an infinitely soft elastic medium. An elasto-viscoplastic law modelled the mechanical behaviour of
each ice crystal. The elasticity tensor was assumed isotropic transverse in the crystal frame and defined according to Gammon
et al. (1983). The viscoplastic part follows the crystal plasticity law proposed by Castelnau et al. (2008); Lebensohn et al.
(2009); Suquet et al. (2012). Ice crystals deform through slip on three soft basal systems, three hard prismatic systems, and six
hard pyramidal systems (Montagnat et al., 2014). Each slip system $(k)$ is governed by a Norton-type flow law, characterised
by a prefactor, the reference shear rate $\dot{\gamma}_0^{(k)}$, a stress exponent $n^{(k)}$, and a critical shear stress $\tau_0^{(k)}$. Different parametrizations
of the slip systems have been proposed in the literature for different purposes: to investigate the viscoplastic development
of texture (Castelnau et al., 1996), to model the viscoplastic behaviour and heterogeneous intracrystalline deformation of
columnar ice polycrystals (Lebensohn et al., 2009), or to simulate the transient regime (Suquet et al., 2012; Castelnau et al.,



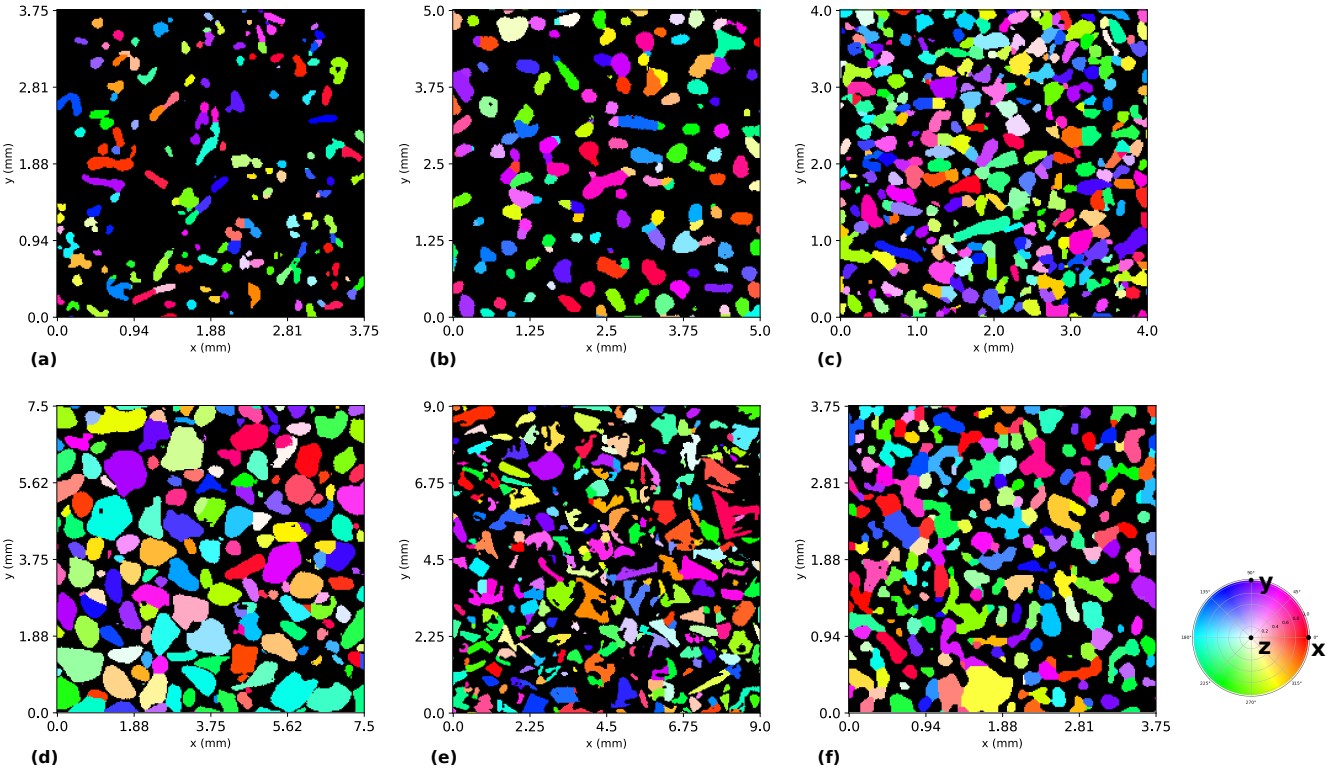

**Figure 2.** Example of the C-axis orientation of different samples: **(a)** PP, **(b)** RG, **(c)** DF, **(d)** RGlr, **(e)** DH, **(f)** MM. Each colour represents a specific c-axis orientation, as indicated by the accompanying colour wheel. The black background corresponds to the original air pores within the snow microstructure. Note that the simulation volumes are 3D. Here only 2D slices are shown.

2008). However, as shown by Védrine et al. (2025), a key point for modelling snow behaviour is to set the exponent for the basal system equal to 2, as extensively experimentally observed (Duval et al., 1983; Schulson and Duval, 2009). We therefore used the stationary parametrization of Suquet et al. (2012), calibrated to reproduce the viscoplastic behaviour of dense polycrystalline ice ($\sigma_0(\Phi = 1) = 272$ MPa and $n_{suquet} = 2.94$) as proposed by Budd and Jacka (1989) and Castelnau et al. (1996) at $T_0 = 263$ K. To account for temperature effects, we used the Arrhenius relation proposed by Mellor and Testa (1969a), i.e., $\dot{\gamma}_0^{(k)}(T) = \dot{\gamma}_0^{(k)}(T_0)A(T)$ with

$$A(T) = \exp\left[-\frac{Q}{R}\left(\frac{1}{T} - \frac{1}{T_0}\right)\right], \tag{3}$$

where $Q = 69.1\,\mathrm{kJ\,mol^{-1}}$ is the activation energy of polycrystalline ice (Mellor and Testa, 1969b), $R = 8.3\,\mathrm{J\,mol^{-1}K^{-1}}$ is the universal gas constant, $T$ is the absolute temperature (in kelvin), and $T_0 = 263$ K denotes the reference temperature at which the parameters are defined.

For the boundary conditions, we prescribed vertical uniaxial compression with periodic boundary conditions (Fig. 3). We imposed a positive average vertical strain rate $\dot{\varepsilon}_{zz} > 0$ and zero average lateral stresses, $\sigma_{xx} = \sigma_{yy} = 0$, and zero average shear





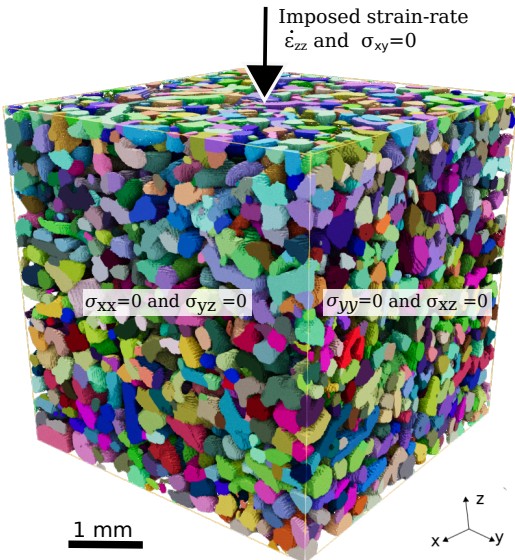

**Figure 3.** Numerical boundary conditions: uniaxial compression test with imposed strain rate.

stresses, $\sigma_{xy} = \sigma_{yz} = \sigma_{xz} = 0$. Here, the macroscopic strain and stress tensors are defined as volume average on the sample: $\varepsilon = \frac{1}{|V|} \int_V \varepsilon(x)\, \mathrm{d}V$ and $\sigma = \frac{1}{|V|} \int_V \sigma(x)\, \mathrm{d}V$, respectively (Hill, 1963). These boundary conditions mimic the conditions of snowpack densification, where gravity is the primary driving force and the influence of confining pressure is negligible (Wautier et al., 2017). We measured the resulting average vertical stress $\sigma_{zz}$ from the simulations.

The simulations were conducted using the Fast Fourier Transform (FFT)-based solver AMITEX_FFTP (Gelebart et al., 2020), using the small perturbation assumption. The crystal plasticity model was implemented using the MFront code generator (Helfer et al., 2015). The numerical integration of crystal plasticity is computationally expensive, but the solver benefits from the MPI implementation.

### 2.1.3 Homogenized model

The mechanical behaviour of heterogeneous materials, such as snow, can often be modelled by replacing the material with an equivalent homogeneous medium. This approximation, also know as homogenization, requires that the characteristic length scale of the material microstructure is small relative to the macroscopic scale of interest (Dormieux and Bourgeois, 2002; Auriault et al., 2010). In the case of snow, this separation of scales is typically satisfied, which allows for the macroscopic mechanical behaviour to be deduced from the mesoscopic structure obtained through X-ray tomography. Previous studies have shown that snow samples on the order of a few millimeters can be considered representative elementary volumes (REVs) for modelling the mechanical behaviour of snow (Wautier et al., 2015; Srivastava et al., 2016; Védrine et al., 2024; Sundu et al., 2024a).



Figure 3 illustrates the uniaxial compression of the exemplary microstructure shown in Fig. 3, under strain rates ranging

from $10^{-7}$ to $10^{-5}\,\mathrm{s}^{-1}$. We observe three different regimes of the stress-strain response (Fig. 4). First, the axial stress $\sigma_{\mathrm{zz}}$ increases linearly with strain in an elastic regime. Then, $\sigma_{\mathrm{zz}}$ deviates from the elasticity, and the viscoplasticity is progressively activated. After this transient regime, $\sigma_{\mathrm{zz}}$ eventually reaches a constant value, the yield stress $\sigma_Y$, where the microstructure flows perfectly (stationary creep). The yield stress depends on the microstructure and the applied strain rate. The permanent viscoplastic regime is reached after a typical strain of 0.5%. Consequently, we conduct simulations up to 1% strain to determine

$\sigma_Y$.

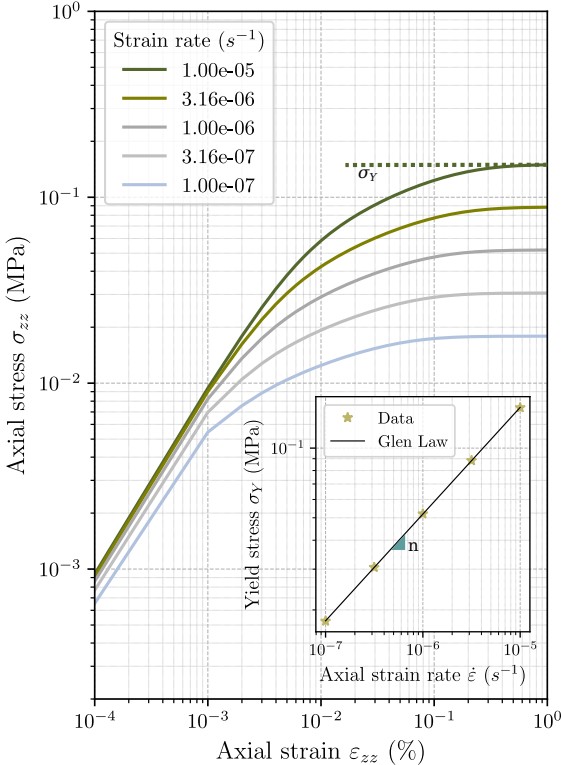

**Figure 4.** Evolution of the vertical stress $\sigma_{zz}$ as the function of the strain $\epsilon_{zz}$ for different strain rates. Inset: Evolution of axial yield stress rate as a function of the imposed strain rate. The tested sample is composed of Decomposing and Fragmented snow particles (DF) with a solid fraction $\Phi = 0.48$, and is shown in Figure 3.

The evolution of yield stress with strain rate follows a power-law over a wide range of strain, encompassing typical strain rates for snow settlement (Fig. 4 inset). Consequently, we can model the constitutive behaviour using the power-law of Eq. 1 by fitting two parameters $n$ and $\sigma_0$. The sensitivity to temperature implemented on the microscopic slip systems, $A(T)$ (Eq. 3), directly scales out at the macroscopic scale. In other words, the microstructure-based simulations yields the following model:

$$\dot{\varepsilon} = \dot{\varepsilon}_0(T_0)A(T)\left(\frac{\sigma}{\sigma_0}\right)^n \tag{4}$$





with $n$ and $\sigma_0$ two fitting parameters depending on the tested microstructure.

## 2.2 Revisiting previous viscoplastic tests

In this section, we describe how to compare previously published viscoplastic tests and gather them into a common framework.

To characterize the viscoplastic behaviour of snow, two types of mechanical tests are commonly employed: load-controlled
tests (creep tests) (Mellor and Testa, 1969b; Desrues et al., 1980; Bartelt and Christen, 1999; Kirchner et al., 2001; Scapozza and Bartelt, 2003b, a; Moos et al., 2003) and strain rate controlled tests (Scapozza and Bartelt, 2003a; Bernard, 2023). We selected experimental data from such tests, where the initial density, temperature, stress, and strain rate were measured during the experiments. To focus exclusively on the viscoplastic behaviour, only tests with a strain rate lower than $4 \times 10^{-4}$ s$^{-1}$ were selected (Kirchner et al., 2001; Barraclough et al., 2017; Löwe et al., 2020; Bernard, 2023). Our dataset comprises 92
measurements obtained from strain rate-controlled tests and 86 measurements obtained from stress-controlled tests (Scapozza and Bartelt, 2003b; Theile et al., 2011; Schleef et al., 2014b; Bernard, 2023; Bernard et al., 2023). One measurement comprises the measured strain rate and stress $(\dot{\epsilon}, \sigma)$ and essential information such as initial ice volume fraction $\Phi$, temperature $T$ and loading conditions (load or strain-controlled). The data is summed up in Table 1 and fully available in the supplementary material (Védrine and Hagenmuller, 2025).

**Table 1.** Characteristic of the experimental data of snow creep used.

| References | Snow type | Temperature (°C) | Density (kg m$^{-3}$) | Strain rate (s$^{-1}$) | Stress (kPa) |
|---|---|---|---|---|---|
| Scapozza and Bartelt (2003b) | RG | [-18.9, -2.1] | [205, 429] | $1.1 \times 10^{-6}$, $4.2 \times 10^{-6}$, $1.1 \times 10^{-5}$, $4.4 \times 10^{-5}$ | [1.7, 258] |
| Theile et al. (2011) S1 S2 | RG | -11 | [205, 345] | $[10^{-7}, 10^{-6}]$ | [2, 10] |
| Schleef et al. (2014b) | PP | [-18, -3] | [ 55, 120] | $[10^{-6}, 10^{-5}]$ | 0.133, 0.215, 0.318 |
| Bernard (2023) V1 | DF/RG | -18.5±0.5 | [218, 467] | $1.8 \times 10^{-6}$ | [0, 300] |
| Bernard (2023) V3 V4 | DF/RG | -18.5±0.5 | [225, 420] | $3.1 \times 10^{-5}$ | [0, 300] |
| Bernard et al. (2023) | DF | -8±0.5 | [236, 260] | $[10^{-8}, 10^{-7}]$ | 2.1 |

The type of snow is defined according to Fierz et al. (2009). The numbers shown in brackets, e.g., $[a,\ b]$, mean that the parameter ranges between, e.g., $a$ and $b$. Details can be found in the associated references.

It is impossible to derive two parameters, namely the reference stress $\sigma_0$ and stress exponent $n$, from a individual measurement points $(\dot{\epsilon}, \sigma)$. Moreover, the data points cannot be directly compared since they were obtained at various temperatures in the range [-18, -2]°C. However, given a certain $n$ in the range [1, 4] and assuming that the Arrhenius temperature scaling



applies (Eq. 3), we rescale the data at a constant temperature of -10°C and we can calculate $\sigma_0(n)$ as:

$$\sigma_0(n) = \sigma \times \left( \frac{\dot{\epsilon}}{\dot{\varepsilon}_0(T_0)A(T)} \right)^{-\frac{1}{n}}. \tag{5}$$

If the power-law is well-formed, $\sigma_0$ and $n$ do not depend on the applied stress or strain rate but solely depend on the sample microstructure. Therefore, for a given microstructure, knowing the correct stress exponent $n$ is equivalent to obtain the same reference stress regardless of the test conducted (loading level or test type). In other words, finding the appropriate constitutive law is equivalent to identifying the exponent $n$ yielding the smallest scatter (defined as the relative standard deviation) of $\sigma_0(n)$ for different experimental tests but similar microstructures.

In this study, we proposed a simplified but pragmatic application of this procedure by considering the solid fraction $\Phi$ as the sole characteristic of the microstructure. Indeed, the characterization of microstructures from data collected in the literature is primarily limited to density. Therefore, we assumed $\sigma_0 = \sigma_0(n, \Phi)$ and $n = n(\Phi)$. Consequently, the exponent $n(\Phi)$ is the one that minimizes the scatter $\sigma_0(n, \Phi)$ for a given narrow range of $\Phi$ across multiple experiments. We will use this conceptual procedure to derive the stress exponent from the presented experimental data (Sect. 3.2). This problem can be formulated as a

minimization problem:

$$\mathcal{L}(\text{model parameters}) = \frac{1}{N_{\text{dataset}}} \sum_{i \in \text{dataset}} \frac{(\tilde{\sigma}_0(\Phi_i) - \sigma_{0_i})^2}{\sigma_{0_i}^2} \tag{6}$$

where $\sigma_{0_i}$ and $\tilde{\sigma}_0$ denote the measured and model-predicted reference stress, respectively.

### 2.3    Snow settlement in detailed snowpack models

The settlement laws currently implemented in snowpack models such as *Crocus* (Brun et al., 1992; Vionnet et al., 2012) and

*Snowpack* (Lehning et al., 2002; Simson et al., 2021) (for vertical stresses lower than 0.4 MPa) are linear. For a given snow layer of thickness $D$, the vertical compaction under an overburden stress $\sigma$ is expressed as:

$$\frac{\mathrm{d}D}{D} = -\frac{\sigma}{\eta} \, \mathrm{d}t \tag{7}$$

where $\mathrm{d}t$ is the model time step and $\eta$ (in MPa·s) denotes the compaction viscosity. The compaction viscosity is typically parametrised as a function of temperature $T$ (in °C) and density $\rho$ (in kg·m$^{-3}$).

In the *Crocus* model (Vionnet et al., 2012), for dry, non-faceted snow, the compaction viscosity follows the formulation of Brun et al. (1992) (hereafter referred to as BR92), derived from the work of Navarre (1975):

$$\eta_{\text{BR92}} = 4\eta_0 \frac{\rho}{c_\eta} \exp(-a_\eta T + b_\eta \rho) \tag{8}$$

where $\eta_0 = 7.62237$ MPa, $a_\eta = 0.1$ C$^{\circ -1}$, $b_\eta = 0.023$ m$^3$kg$^{-1}$, and $c_\eta = 250$ kg·m$^{-3}$.

The ensemble version of the French snowpack model *Crocus* (ESCROC) (Lafaysse et al., 2017) also includes the viscosity

parameterisation of Teufelsbauer (2011) (hereafter referred to as T11), derived from the observations of Narita (1984) and





corrected for low-density snow using the laboratory-based model of Abe (2001):

$$\eta_{\text{T11}} = 5 \cdot 10^{-8} \, \rho^{(-0.0371T+4.4)} \left(10^{-4} \exp(0.018\rho) + 1\right) \tag{9}$$

In the *Snowpack* model (Lehning et al., 2002), for dry new snow (with dendricity greater than zero), the compaction viscosity follows the empirical formulation of Kojima (1975) (hereafter referred to as K75), derived from field observations of the natural 230 settlement of seasonal snow:

$$\eta_{\text{K75}} = 7 \cdot 10^{-9} \, \rho^{(4.75 - T/40)} \tag{10}$$

Although viscosity is a material parameter that characterises deformation in a specific direction, it is commonly assumed that for densities below $450 \, \text{kg·m}^{-3}$, boundary conditions have a negligible effect (Wautier et al., 2017). Consequently, free and confined boundary conditions are not generally distinguished in these models.

## 235 3 Results and discussion

### 3.1 Microstructure-based model

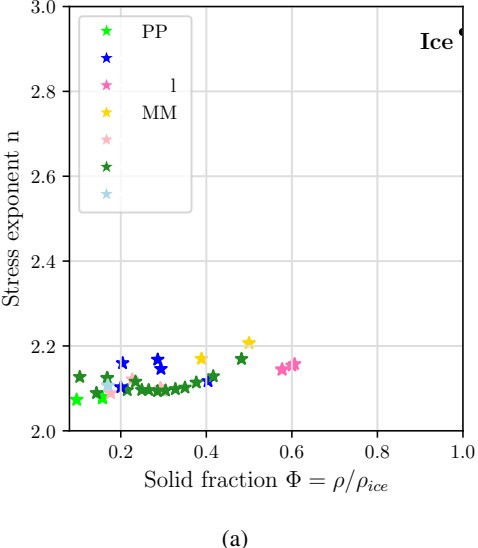
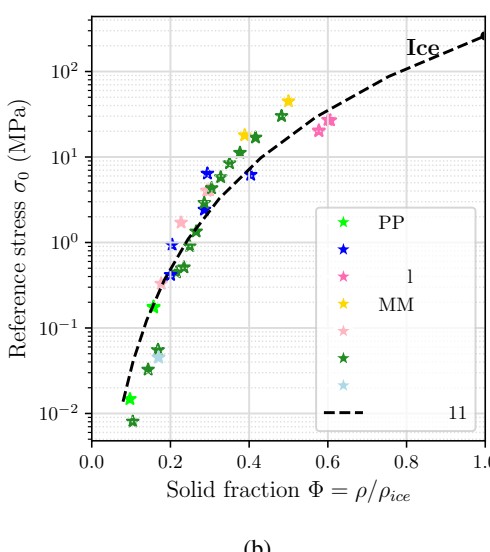

(a)                                                      (b)

**Figure 5.** Parameters of the homogenized model: **(a)** stress exponent $n$ and **(b)** reference stress $\sigma_0$ as a function of the solid fraction $\Phi$.

Figure 5 presents the results from the microstructure-based simulations conducted on the 37 tested samples.

Figure 5a shows the evolution of the stress exponent $n$ with the solid fraction $\Phi$. The stress exponent ranges between 2.1 and 2.2 for the considered microstructures with $\Phi$ between 0.1 and 0.6. These values are lower than the exponent of dense





ice ($n_{\mathrm{ice_{suquet}}} = 2.94$) and are close to the basal slip exponent ($n_{basal} = 2$). As shown by Védrine et al. (2025) on synthetic

polycrystals, the homogenized exponent depends on the activity of the slip systems. Specifically, as the incompatibility of

deformation between crystals decreases, the activity of the softer slip systems increases and the stress exponent $n$ tends towards

2. We do not observe a large sensitivity of $n$ to $\Phi$, it is rather constant with an average value of $n_{\mu CT}$ =2.15. Indeed, the surface

ratio between the inter-crystalline surfaces and the free surfaces is almost constant and remains small (approximately 0.1, see

supplementary material (Védrine and Hagenmuller, 2025) and Figure B1), which yields a low frustration of the basal slip

systems (Védrine et al., 2025). Consequently, we do not observe an increase of $n$ with $\Phi$.

The reference stress $\sigma_0$ increases with the solid fraction $\Phi$ (Fig. 5b). Indeed, when the solid fraction increases, the stress

supported by the ice skeleton (for a given macroscopic stress $\sigma$ averaged in air and ice) decreases, leading to an increase of the

reference stress. Following Védrine et al. (2025), we fitted the evolution of $\sigma_0$ with $\Phi$ with a power-law:

$$\sigma_0 = \sigma_0(\Phi = 1)\Phi_r^m. \tag{11}$$

with $\Phi_r$ the rescaled solid fraction (Green, 1998; Holman and Leuenberger, 1988; Stauffer and Aharony, 2018) defined as:

$$\Phi_r = \frac{\Phi - \Phi_t}{1 - \Phi_t} \tag{12}$$

The quantity $\sigma_0(\Phi = 1) = 272$ MPa represents the reference stress of dense polycrystalline ice, $m$ is the solid fraction expo-

nent, and $\Phi_t$ is the percolation threshold. By fitting the data obtained through homogenization, we found $\Phi_t \approx 0$, $m = 3.96$, and

a scatter of $\mathcal{L} = 1.68$ (Eq. 6) around this model. The very low value for $\Phi_t$ suggests that the snow microstructure can maintain

connectivity (i.e., percolate) even at very low solid fractions. The solid fraction sensitivity $m$ relates to the heterogeneity of

viscoplastic deformations within the sample. More generally, the value of $m$ can be computed for different snow properties,

such a thermal conductivity, Young's modulus, and strength. It serves as an indicator of the heterogeneity of the local driving

processes: higher values of $m$ reflects higher heterogeneity (modulated by both the process type and microstructure) (Bruno

et al., 2011). The fitted value of $m = 3.96$ for snow viscoplasticity is similar to the ones computed for other snow physical

properties. For thermal conductivity, $m$ typically ranges between 1 and 2 (Calonne et al., 2011, 2019). For the elastic modulus,

$m$ usually falls between 2.5 and 3.5 (Gaume et al., 2017; Sundu et al., 2024a). For strength, Ritter et al. (2020) found values

of 3.59 in compression and 3.09 in tension. Overall, the pseudo-linear viscosity $\eta$ scales with $\phi^{-nm}$ (Eq. 2). This extreme

sensitivity to solid fraction emphasizes the challenges in characterizing the viscoplastic behaviour of snow. For example, a 5%

relative error in measuring the solid fraction, which is common (e.g. Proksch et al., 2015; Hagenmuller et al., 2016) leads to a

relative error of about 20% error in the reference stress $\sigma_0$ and a relative error of around 50% for the compaction viscosity $\eta$.

The scatter observed in the loss value, $\mathcal{L} = 1.68$, can be attributed to the fact that only density is considered as a predic-

tive indicator. However, it is evident that RGlr exhibits a reference stress approximately four times lower than that of other

snow types. Similarly, the DF series studied by Bernard (2023) transitions from being underestimated to overestimated by the

model, suggesting that under loading, the microstructure tends to optimize itself. These two examples suggest the influence of

additional geometric parameters, such as bond size (Hagenmuller et al., 2014a) which are currently not accounted for in the

model.



## 3.2 Experimental data-driven model

To reinterpret previous experimental data, our idea was to determine the value of $n$ that minimizes the scatter of the reference
stress $\sigma_0$ for "similar" solid fractions $\Phi$ across different experiments (see Sect. 3.1 and Eq. 11). The microstructure-based
simulations showed that the reference stress is likely to be very sensitive to $\Phi$: $\sigma_0 \sim \Phi^{3.96}$. It is thus impossible from the
collected data (178 points) to gather enough points within a narrow range of $[\Phi, \Phi + \Delta\Phi]$ with $\Delta\Phi \ll 1$ to accurately quantify
the scatter of $\sigma_0$ and the value $n(\Phi)$. However, the microstructure-based simulations also revealed that $n$ is relatively constant
for snow and $\sigma_0$ follows a power-law of $\Phi$. We use these two results from the microstructure-based simulations as additional
assumptions to reinterpret previous experimental data. In other words, we want to find a constant $n$ that minimizes the relative
scatter $\mathcal{L}(n)$ (defined as the relative standard deviation, see Eq. 6) of $\sigma_0$ around a power-law of $\Phi$ of the form $\tilde{\sigma}_0(n) = \sigma_0(\Phi = 1)\Phi_r^m$, with $\Phi_r$ the rescaled solid fraction (Eq. 12) and, $m$ and $\Phi_t$ two optimal fitting parameters depending on $n$.

We tested this procedure with three different values for the stress exponent: $n = 1$, $n = 3$, and $n = 2.15$ (Fig. 6):

– *n=1* (Fig. 6a). When the viscoplastic behaviour is assumed to be linear, we observe significant dispersion in the reference
stress $\sigma_0$ ($\mathcal{L} = 0.87$), particularly at solid fractions exceeding 0.2. Notably, the dataset divides into two distinct groups:
the tests with imposed strain rates are located in the lower part of the scatter plot, while the creep tests are located in the
upper part. This dispersion and the separation into two groups indicate that a linear model fails to adequately reproduce
the variety of experimental data. Figure 6 also includes the linear models from Brun et al. (1992), Teufelsbauer (2011)
and Kojima (1975) at a temperature of $-10°C$. The Br92 model overestimates all experimental reference stresses and
thus appears too fluid. Moreover, it reaches the viscosity of ice for an ice volume fraction of about 0.5. The T11 model
also tends to overestimate the reference stress but aligns more closely with the experimental values for low-density snow
measured by (Schleef et al., 2014b). Finally, the K75 model reproduces the experiments of Schleef et al. (2014b) with
good fidelity and lies near the centre of the experimental data set.

– *n=3* (Fig. 6b). The case $n = 3$ corresponds to the exponent of dense polycrystalline ice. It is commonly assumed that
this exponent of the dense material applies to its porous counterpart, snow (Kirchner et al., 2001; Wautier et al., 2017;
Fourteau et al., 2023). In this case, the dispersion decreases to $\mathcal{L} = 0.19$. This decrease is particularly visible for snow
with solid fractions above 0.2. However, the values of $\sigma_0$ are still divided into two groups depending on the type of test.

– *n=2.15* (Fig. 6c). Finally, when we assume that the stress exponent is equal to $n_{\mu CT} = 2.15$, the dispersion is reduced to
$\mathcal{L} = 0.14$, with optimized parameters $\Phi_t = 0.025$ and $m = 3.195$. Moreover, unlike the case with $n = 3$, the experiments
with imposed stress and strain rates are superposed, which indicates an appropriate underlying model.

Figure C1 shows the evolution of the dispersion $\mathcal{L}$ as a function of $n$. $\mathcal{L}$ decreases when $n$ increases from 1 to 2.35. For $n$
in [2.35, 2.59], $\mathcal{L}$ remains nearly constant with a weak minimum at $n = 2.53$. For $n > 2.59$, $\mathcal{L}$ increases with $n$. Interestingly,
the minimum dispersion error is obtained for a value of $n$ close to the one obtained from microstructure-based simulations.
This parametrization significantly improves the modelling of the viscoplastic behaviour of snow. The exponent $n = 2.15$ is
consistent with the value derived by Scapozza and Bartelt (2003b) and Schleef et al. (2014b) on samples with low density





**Figure 6.** Experimental data on the evolution of the reference stress as a function of solid fraction, recalculated under various parametrisations of $n$ at a temperature of -10°C.



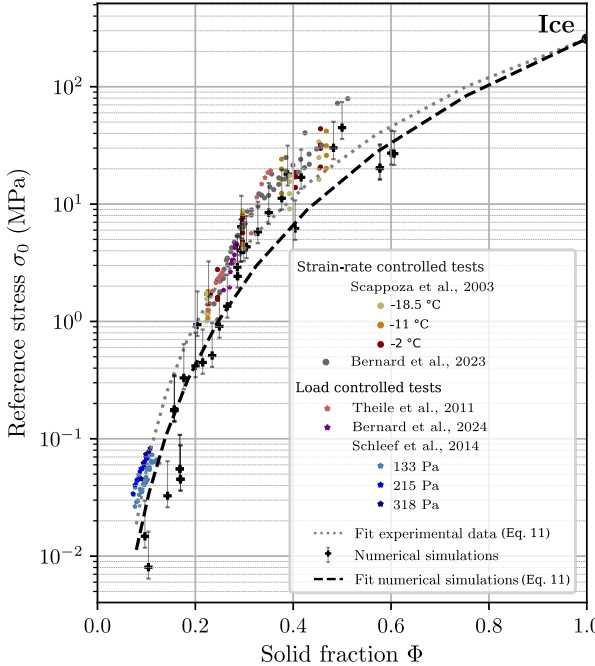

**Figure 7.** Comparison of the microstructure-based model and experimental data regarding the evolution of the reference stress as a function of solid fraction, calculated with the assumption $n = 2.15$ at a temperature of -10°C.

($\Phi < 0.2$). This agreement supports the idea that basal glide is the dominant mechanism at low density. Furthermore, this suggests that the reduction of the stress exponent from dense ice to snow does not require introducing grain boundary sliding as a new mechanism, in contrast to the intuition of Alley (1987) and Goldsby and Kohlstedt (1997).

The residual error obtained, $\mathcal{L}(n = 2.15) = 0.15$ is not null and may be attributed to model error and experimental measure-
ment noise. First, the microstructure is simplified here to its solid fraction, $\Phi$. Our model cannot account for other geometrical properties, such as bond size (Hagenmuller et al., 2014a) or connectivity (Schleef et al., 2014b; Schöttner et al., 2025), or the inter-crystalline surface area, which affects the frustration of the basal glide (Védrine et al., 2025). Indeed, the residual error computed on the microstructure-based simulations is also not null ($\mathcal{L} = 1.68$), which tends to show that the variety of microstructural patterns for a given $\Phi$ could be enriched by other microstructural descriptors. Last, we showed that $\sigma_0$ varies
like $\Phi^m$ with $m \simeq 3$. Therefore, a small error in measuring $\Phi$, or a small heterogeneity of density in the prepared sample, may significantly affect the measured points and subsequent analysis. Second, the scaling to a constant temperature of -10°C might be a source of uncertainty. For instance, the residual dispersion between appears more pronounced at solid fractions of 0.4 and 0.5 (Fig. 6c), corresponding to tests conducted by the same operator (Scapozza and Bartelt, 2003a) under the same conditions except different temperatures ($[\sim -18.5, \sim -11, \sim -2]$). Likely, the Arrhenius scaling around -10°C (Eq. 3) does not apply
to temperatures close to 0°C and below -15°C where different deformation mechanisms may take place (Mellor and Testa, 1969b).





Figure 7 illustrates the reference stress $\sigma_0$ computed from microstructure-based simulations alongside the experimental re-analysis, both with $n = 2.15$. We observe that the microstructure-based reference stress deviates from the data-driven reference stress by a factor of 2. This difference indicates that the viscoplastic behaviour simulated by considering only intracrystalline

deformations might be too fluid. Thus, incorporating an additional deformation mechanism, such as grain boundary sliding, would likely render the behaviour even more fluid. One possible explanation for this lower viscosity may be the assumption that a geometric grain corresponds to a crystal (Védrine et al., 2025). Finally, both fits to experimental data and microstructure-based simulations tend to underestimate viscosity for a solid fraction around 0.4, suggesting a potential evolution of the morphology parameter $m$, which is here assume constant for $\Phi$ in the range $[\Phi_t, 1]$. Indeed, Coble (1961) showed that for $\Phi$ close to 1, i.e.

diluted pores in an ice matrix, the solid fraction exponent $m$ is closer to 1.

### 3.3   Difference between the data-driven non-linear model and the settlement law implemented in Crocus

In this section, we compare the predictions of the detailed snowpack model Crocus (referred to as Br92) to those of our data-driven non-linear model (referred to as V25):

$$\dot{\epsilon}(\sigma, T) = e^{\frac{-Q}{R}\left(\frac{1}{T} - \frac{1}{T_0}\right)} \left(\frac{\sigma}{\sigma_{0(\Phi=1)}\Phi_r^m}\right)^n, \tag{13}$$

with $Q = 69.1\,\mathrm{kJ\,mol^{-1}}$, $T_0 = 263$ K, $n = 2.15$, $m = 3.195$ and $\Phi_r$ the rescaled solid fraction (Eq. 12) with $\Phi_t = 0.025$. The evaluation is focused on stress - solid fraction $(\sigma, \Phi)$ couples, which are realistic for natural snowpacks. We recall that the temporal evolution of the thickness $h$ of a layer subjected to an overburden stress $\sigma$ is related to strain rate as $\dot{\epsilon} = \dot{h}/h$.

Figure 8 shows the relationship between the applied stress and the solid fraction for layers measured in natural seasonal snowpacks at different locations: Cambridge Bay (High-Arctic, Canada, winter 2023-2024, ongoing ERC IVORI campaign),

Col de Porte (French Alps, 1325 m, winters 1993-1994 to 2017-2018, Lejeune et al. (2019)), and Weissflüjoch (Swiss Alps, 2536 m, winter 2015-2016, Calonne et al. (2020)). The applied stress was calculated as the overburden at mid-layer. It mainly ranges between 0.1 kPa and 4 kPa and generally increases with the solid fraction. The two alpine sites exhibit larger stresses for a given solid fraction compared to the Arctic site. In the dry arctic region, densification is primarily driven by wind erosion-deposition, whereas at the alpine sites, gravity-driven creep is the dominant settlement mechanism (Woolley et al., 2024).

The predictions of V25 and Br92 are compared in the stress - solid fraction diagram shown in Figure 8. The Br92 model exhibits a lower stress sensitivity due to its linear formulation. Consequently, our parametrisation V25 predicts higher strain rates for large loads and lower strain rates for small loads, compared to Br92. Notably, the two models yield equivalent results for the solid fraction–stress couples most commonly observed at the Col de Porte, where the Br92 model has been extensively calibrated (Navarre, 1975; Brun et al., 1992). This agreement supports the validity of the V25 model. The linear model performs

well when the applied stress does not vary much beyond its calibration points or when the stress correlates with density. In this latter case, the non-linearity can be artificially introduced through the dependence of viscosity $\eta$ on solid fraction $\Phi$. However, substantial discrepancies arise when moving beyond these "standard" conditions: Br92 over-estimates the strain rate for $\Phi > 0.08$ and stresses smaller than those observed at col de Porte, Br92 under-estimates the strain rate for very low density samples ($\Phi < 0.08$) and for stresses higher than the "standard" ones.




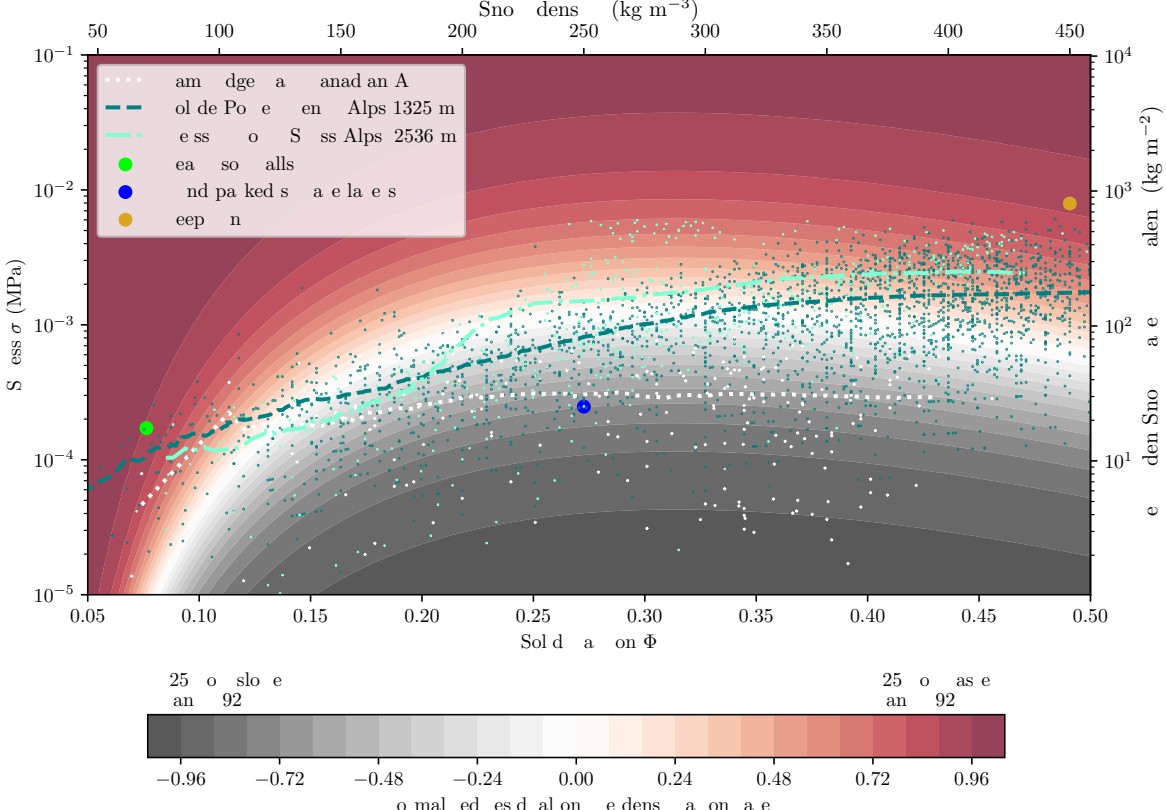

**Figure 8.** Normalized residual $R = \frac{\dot{\epsilon}_{V25} - \dot{\epsilon}_{Br92}}{\dot{\epsilon}_{Br92} + \dot{\epsilon}_{V25}}$ between the V25 and Br92 model for varying solid fractions and stresses. The stress is converted into its snow water equivalent.

We illustrated this discrepancy on three exemplary cases, which are out of the common conditions encountered at Col de Porte:

- *Heavy snowfalls*. We consider the bottom of a 25 cm thick layer of snow with a density of 70 kg m$^{-3}$ ($\Phi = 0.076$, $\sigma = 0.17$ kPa, see green point on Fig. 8). This scenario typically corresponds to a layer formed during a heavy snowfall. During the first day, the V25 model shows densification at a rate up to 33 times that predicted by the Br92 model (Fig. 9).

Interestingly, the two models converge after approximately 14 days, as the increase in density leads to a corresponding rise in viscosity. This finding aligns with field observations, which often indicate that densification is frequently underestimated in the immediate aftermath of snowfall (Lundy et al., 2001; Steinkogler et al., 2009; Wever et al., 2015) although it has minimal impact in the long term. Higher strain rates thus naturally arise on recent snow in the V25 model without introducing a distinct role for curvature-driven metamorphism (Chen et al., 2019).





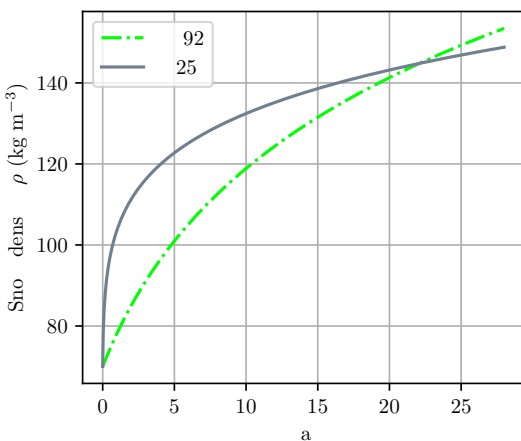

**Figure 9.** Evolution of the density at the bottom of 25 cm layer of new snow at 70 kg m$^{-3}$ with the model Br92 et V25 at -10°C.

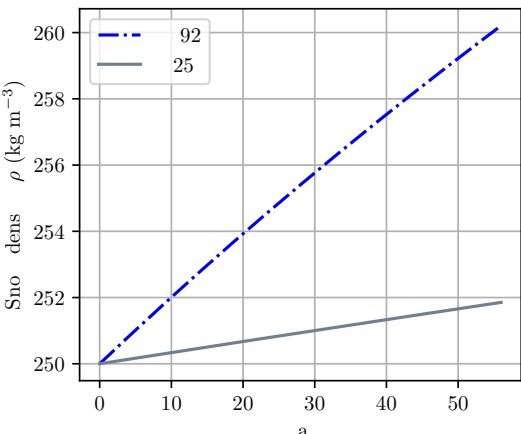

**Figure 10.** Evolution of the density at the bottom of 10 cm thick layer of snow at 250 kg m$^{-3}$ with the model Br92 et V25 at -10°C.

– *Wind-packed surface layers*. We consider the bottom of a 10 cm thick layer of snow with a density of 250 kg m$^{-3}$ ($\Phi = 0.22$, $\sigma = 0.25$ kPa, see blue point on Fig. 8). This layer typically forms at the snowpack surface during a wind event. The V25 model predicts a densification rate about six times slower in the first few days compared to the Br92 model (see Fig. 10). This result aligns with observations of limited densification of basal layers in Arctic snowpacks—behaviour that is not well captured by existing parametrisations. Nevertheless, our model does not distinguish snow types and cannot
predict the distinct densification rates observed between RG and DH at a given density (Fourteau et al., 2023).

– *Deep firn*. We consider a layer of firn with a density of 450 kg m$^{-3}$ buried beneath 2 m of snow with an average density of 400 kg m$^{-3}$ ($\Phi = 0.49$, $\sigma = 11.9$ kPa, see yellow point on Fig. 8). This layer typically forms on an accumulation zone on an alpine glacier. The V25 model predicts a densification rate that can be up to 30 times faster than that predicted by





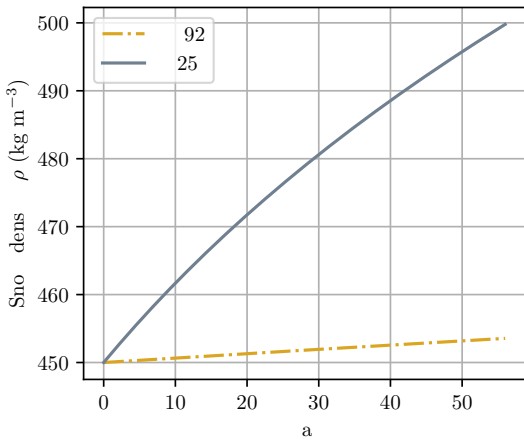

**Figure 11.** Evolution of the density of a layer of firn at 450 kg m$^{-3}$ buried below 2 m of snow with an average density of 400 kg m$^{-3}$) with the model Br92 et V25 at -10°C.

the Br92 model (Fig. 9). This result is consistent with field observations of the densification of dense snow and firn in
glacier accumulation zones (Touzeau et al., 2018; Verjans et al., 2019), where flow rates are frequently underestimated. For instance, Kappelsberger et al. (2024) tuned the viscosity of BR92 by a factor of 1.4 to model polar firn settlement. Nevertheless, to broaden the applicability of the V25 model to firn at densities exceeding 450 kg m$^{-3}$, we will need to investigate the effects of lateral confinement in greater detail, which are negligible only for low-density snow (Wautier et al., 2017).

## 3.4   Intrinsic limitations of the proposed settlement model

We have integrated previously separate approaches into a cohesive, physically grounded framework, allowing us to reconcile results that may initially appear contradictory. However, our framework (Eq. 13) cannot, at this stage, capture known features of snow settlement because it over-simplifies the snow microstructure or the underlying processes. The limitations that would need to be addressed in future work notably comprise:

– *Accounting for morphology descriptors*. Our model simplifies the snow microstructure to its ice fraction $\Phi$. This common approach does not capture second-order effects of the microstructure on its mechanical behaviour (Hagenmuller, 2014). Indeed, we observed scatter of the simulated reference stress at a given density (Fig. 5). We chose not to introduce any additional microstructural descriptors to maintain compatibility with existing experimental data or detailed snowpack models in which the microstructure is very roughly described. However, conducting the microstructure-based simulations
on a larger set of 3D images could help identify key additional descriptors of the microstructure (e.g. Hagenmuller et al., 2014a; Schöttner et al., 2025). These descriptors will need to be predicted by the next generation of snowpack models (e.g. Brondex et al., 2023).



– *Accounting for the inter-crystalline surface area.* Our microstructure-based simulations assume that one grain detected with geometrical criteria (bond ∼ neck) corresponds to one crystal. This assumption is reasonable for rounded grains, but is more questionable for faceted crystals or depth hoar. Védrine et al. (2025) showed that the value of the stress exponent $n$ is sensitive to crystal frustration, which depends on the ratio between the free surface area and the inter-crystalline surface area. Based on our assumption, the ratio of inter-crystalline to free surface area was found to be nearly constant and small (approximately 0.1), implying a low degree of basal slip-system frustration. We believe that this inter-crystalline surface area plays a key role in the observed differences in settlement rates between rounded grains and depth hoar (Kojima, 1967; Armstrong, 1980; Sturm and Holmgren, 1998). In particular, we intuit that, for a given density, depth hoar is characterised by a higher frustration compared to rounded grains, which yields slower densification. This lower settlement rate of depth hoar is not anecdotal. It is crucial in the formation of persistent weak layers, a prerequisite for slab avalanches, or in capturing the thermal properties of widespread tundra snowpacks (Fourteau et al., 2023). Due to the lack of knowledge about the exact crystalline structure, we cannot currently test this intuition. This idea thus warrants further numerical investigations using the proposed 3D model associated with explicit crystal measurements via diffraction contrast tomography (Granger et al., 2021).

– *Refining the temperature sensitivity (Arrhenius law).* We assumed that the temperature dependence of the homogenised Glen's law follows an Arrhenius relationship with a single activation energy applied uniformly across all microstructures and temperatures (Eq. 3). However, the activation energy in dense polycrystalline ice varies with temperature (Mellor and Testa, 1969b), and each crystallographic slip system has its own activation energy (Mellor and Testa, 1969b; Ramseier, 1975; Goldsby and Kohlstedt, 2001), affecting the relative activity of different systems. Consequently, a temperature dependence of the stress exponent is to be expected, as noted by Scapozza and Bartelt (2003b). Besides, explicit measurements of the activation energy for snow are seldom reported (Delmas, 2013; Schleef et al., 2014b) and can be affected by microstructural changes between tests or during a single test. Knowledge of the temperature sensitivity of the driving deformation mechanisms would help develop a unified settlement law applicable over a broad range of temperatures.

– *Extending the approach to a 3D constitutive model.* We evaluated our model on snow with a solid fraction, $\Phi \leq 0.5$, but it could be extended to larger solid fractions, e.g., to capture the transition from snow to glacier ice. However, our model is one-dimensional and assumes uniaxial compression with zero average horizontal stresses. Snow, firn or ice in natural conditions may also be subjected to horizontal stresses. For snow with a low solid fraction, these stresses can generally be neglected (Wautier et al., 2017), but this is not the case any more for firn or porous ice. Therefore, it would be necessary to develop a complete 3D constitutive model to capture this effect.

## 4 Conclusions and perspectives

We used three different sources of information to understand the viscoplastic behaviour of snow:





– *Microstructure-based simulations*. We simulated the viscoplastic behaviour of snow based on its three-dimensional microstructure and a crystal-plasticity law for the ice matrix. Our simulations revealed, across a variety of microstructures, that the viscoplastic behaviour of snow followed a power-law $\dot{\epsilon} = \dot{\epsilon}_0 (\sigma/\sigma_0)^n$. The stress exponent $n$ ranged from 2.1 to 2.2. The reference stress $\sigma_0$ was found to be a function of the solid fraction $\Phi$, following the relationship $\sigma_0 \sim \Phi_r^m$. The values for $n$ are consistent both with experimental results for low-density snow (Scapozza and Bartelt, 2003b; Schleef et al., 2014b) and with generic predictions for porous polycrystals (Védrine et al., 2025). These values are close to the stress exponent of the basal dislocation glide, since the ice crystals are poorly constrained by neighbours in snow.

– *Previous viscoplastic tests*. We revisited a dataset comprising 178 measurement points $(\sigma, \dot{\epsilon})$ and found that linear models fail to reproduce the viscoplastic behaviour of snow under various loading conditions. In contrast, using the stress exponent derived from microstructure-based simulations ($n = 2.15$) significantly reduces the variability of the reference stress between independent measurement sets, especially between load-controlled and displacement-controlled tests. Additionally, it was unnecessary to adjust $n$ over a wide range (e.g., [1, 4], which exceeds the range predicted by the microstructure-based model [2, 3]) to consolidate previous datasets. We obtained a relatively simple constitutive relation presumably valid across a broad range of loading conditions (Eq. 13).

– *Detailed snowpack models*. Snowpack models, such as Crocus, assume a linear relation between stress and strain rate ($n = 1$) and are well-calibrated on field data. We showed that in typical natural seasonal alpine snowpacks, the applied stress (or layer depth) is not independent of the ice fraction. Therefore, linear and non-linear models can coincide under these specific conditions. In fact, the linear viscosity $\eta$, a function of solid fraction (which itself is a function of stress), can artificially introduce non-linearity between stress and strain rate. Indeed, we showed that Crocus and our model yield equivalent predictions on conditions typically encountered on alpine sites. However, large deviations appear under "non-standard" conditions, such as high loads on light snow or low loads on dense snow, where our model appears to perform better qualitatively.

We have thus unified previously disparate approaches within a single, physically grounded framework, thereby reconciling results that might initially seem contradictory. This framework provides a robust basis for improving the representation of snow settlement processes in numerical snowpack models.

*Code and data availability.* All materials used in this article (codes, geometric characteristic of the microstructure, numerical values, experimental data etc.) are available at Zenodo repository (Védrine and Hagenmuller, 2025)

## Appendix A:  Constitutive Law: Crystal plasticity model

The total strain rate tensor $\boldsymbol{\varepsilon}$ is decomposed into an elastic part ($\boldsymbol{\varepsilon}^e$) and a viscoplastic part ($\boldsymbol{\varepsilon}^{vp}$), such that: $\boldsymbol{\varepsilon} = \boldsymbol{\varepsilon}^e + \boldsymbol{\varepsilon}^{vp}$.




The elastic strain $\varepsilon^e$ is related to the stress tensor $\boldsymbol{\sigma}$ and the fourth-order stiffness tensor $\mathbf{C}$ by the relation:

$$\boldsymbol{\sigma} = \mathbf{C} : \varepsilon^e. \tag{A1}$$

The elasticity tensor $\mathbf{C}$ is assumed to be isotropic in the crystal reference frame and is defined according to Gammon et al. (1983) as follows: $C_{11} = 13.9\,\text{GPa}, \quad C_{33} = 15.0\,\text{GPa}, \quad C_{44} = 3.0\,\text{GPa}, \quad C_{12} = 7.1\,\text{GPa}, \quad C_{13} = 5.8\,\text{GPa}.$

For the viscoplastic part, at infinitesimal strains, the viscoplastic strain results from slips on a total of $N$ different slip systems:

$$\dot{\boldsymbol{\varepsilon}}^{vp} = \sum_{k=1}^{N} \dot{\gamma}_0^{(k)}(T) \left( \frac{|\tau^{(k)}|}{\tau_0^{(k)}} \right)^{n^{(k)}} \text{sgn}(\tau^{(k)}) \boldsymbol{\mu}^{(k)} \tag{A2}$$

where $n^{(k)}$ is the stress exponent, $\dot{\gamma}_0^{(k)}$ is the reference shear rate, $\tau_0^{(k)}$ is the critical shear stress for slip system $(k)$, and $\boldsymbol{\mu}^{(k)} = \boldsymbol{n}^{(k)} \otimes_s \boldsymbol{b}^{(k)}$ is the Schmid tensor defined by the (symmetric) dyadic product $(\otimes_s)$ of the vector normal to the glide plane $\boldsymbol{n}^{(k)}$ and the Burgers vector $\boldsymbol{b}^{(k)}$ for slip system $(k)$. The critical shear stresses $\tau_0^{(k)}$ are assumed to be constant (no hardening), and the reference shear rate is set to $\dot{\gamma}_0^{(k)} = 1\,\text{s}^{-1}$ at $-10°\text{C}$. The parametrizations of the crystal plasticity model are summarized in Table A1.

**Table A1.** Single crystal ice parameters of crystal plasticity models at $-10\,°\text{C}$ (Suquet et al., 2012).

| Family | Systems | $n^{(k)}$ | $\tau_0^{(k)}$(MPa) |
|---|---|---|---|
| Basal $[0001] < 11\bar{2}0 >$ | 3 | 2 | 35 |
| Prismatic $[01\bar{1}0] < 2\bar{1}\bar{1}0 >$ | 3 | 2.85 | 166 |
| Pyramidal $[11\bar{2}2] < 11\bar{2}\bar{3} >$ | 6 | 4 | 98 |

**Appendix B: Evolution of the surface ratio between the inter-crystalline surfaces and the free surfaces with the microstructure**

The evolution of the surface ratio between inter-crystalline surfaces and free surfaces, denoted as $r$, is plotted as a function of the solid fraction for different microstructures (Figure B1). We observe that $r$ increases only slightly with the solid fraction. A plateau is reached at $r = 0.1$ for solid fractions ranging from 0.2 to 0.45, with only the MM and RGlr snow types deviating
from this trend. This low surface ratio leads to low frustration of the basal slip systems (Védrine et al., 2025).



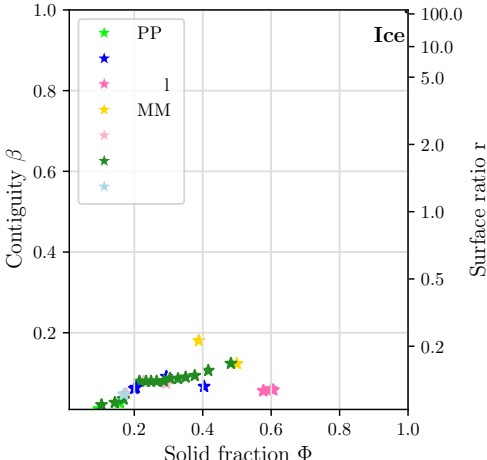

**Figure B1.** Contiguity $\beta$ and surface ratio $r$ as the function of solid fraction $\Phi$. The contiguity Underwood (1970) is defined as $\beta = \frac{r}{1+r}$.

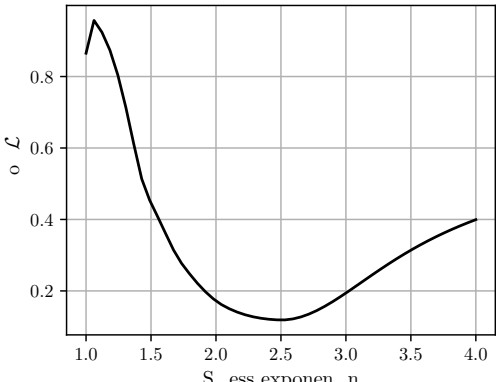

**Figure C1.** Evolution of the dispersion $\mathcal{L}$ as a function of stress exponent $n$ at a temperature of -10°C.

## Appendix C: Evolution of $\mathcal{L}$ as a function of stress exponent $n$.

*Author contributions.* L.V. developed the microstructure-based model, performed the simulations, compiled the data, and created the figures. L.V. and P.H. interpreted the results and wrote the paper. P.H. designed the study.

*Competing interests.* The authors declare that they have no conflict of interest.



*Acknowledgements.* The computations presented in this paper were performed using the GRICAD infrastructure (https://gricad.univ-grenoble-alpes.fr), which is supported by CNRS, University Grenoble Alpes and INRIA. CEN is part of LabEx OSUG@2020 (ANR-15-IDEX-02). We thank the European Research Council (ERC) for funding, under the European Union's Horizon 2020 research and innovation programme (grant agreement no. 949516, IVORI), the Arctic field campaign in Cambridge Bay. The tomography apparatus (TomoCold) was funded by INSU-LEFE, Labex OSUG (Investissements d'avenir–ANR10 LABX56). We thank H. Löwe for fruitful discussions.





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
