# Peer review of "Revisiting snow settlement with microstructural knowledge"

_EGUsphere, 2025_

## Referee Comment (RC1)

November 10, 2025

To the Editors/Authors,

I have reviewed the manuscript *"Revisiting snow settlement with microstructural knowledge"* submitted by Louis Védrine and Pascal Hagenmuller to *The Cryosphere.* Overall, I find this to be a very strong and well-crafted study that makes a meaningful contribution to our understanding of the viscoelastic and microstructural controls on snow settlement. The work combines microstructure-based crystal plasticity modeling with a thoughtful analysis of previous creep data, and the synthesis presented here brings valuable physical clarity to a problem that has long been described in more empirical terms.

The manuscript fits well within the aims and scope of *The Cryosphere.* It meets the journal's criteria for originality, scientific rigor, and clarity, and it provides new insights of practical importance to snowpack and firn modeling communities.

**General Evaluation**

The study is carefully designed and well executed. The modeling approach is technically sound, the connection to experiments is well argued, and the presentation is clear. The results convincingly show why seemingly linear settlement laws appear to work under typical Alpine conditions, while also explaining where and why these relationships may fail.

I have only a few points that I believe should be clarified before publication. These are primarily matters of explanation rather than scientific substance.

**Specific Comments**

1. On the 1% strain and the strain-rate minimum

Around line 175, the authors use 1% strain to determine the steady (minimum) creep rate. This is a reasonable and widely accepted choice: for many materials, including polycrystalline ice, the minimum creep rate occurs at about 1% strain. Jacka (1984) reported the minimum at roughly 0.6%, and Treverrow et al. (2012) found steady conditions near 1%. I suggest the authors briefly note this and include one or two references to show that their cutoff corresponds to the transition from transient to steady-state creep.

2. On the stress exponent and grain boundary sliding

Lines 307–308 state that the observed stress exponent ($n \approx 2$) does not require invoking grain boundary sliding (GBS). This conclusion seems plausible given the microstructure-based modeling. However, previous studies—such as Alley (1987) and Goldsby & Kohlstedt (1997, 2001)—have associated $n \approx 2$ with GBS in fine-grained ice. It would be helpful

to acknowledge that this similarity exists and to explain that the present work offers an alternative physical explanation (based on constrained basal glide in a porous polycrystal) for the same effective exponent. Doing so would better situate the results in the context of existing literature.

3. On crystal orientation and slip systems

The model includes basal, prismatic, and pyramidal slip systems. Because basal slip dominates and is rotationally symmetric about the c-axis, the specific a-axis orientation typically has little effect on mechanical behavior at these stress levels. Still, it would be worth clarifying whether the simulations explicitly included full crystal orientations, or if c-axis randomization alone was used. A short note referencing Schulson & Duval (2009) or Weikusat et al. (2017) would be helpful to reassure readers that this simplification is appropriate.

4. On the word "frustration"

The term "frustration" appears several times and is used correctly in the mechanical sense—referring to geometric or kinematic incompatibility between neighboring grains that limits basal glide. For readers unfamiliar with materials-science terminology, I suggest adding a brief clarification at its first occurrence (e.g., "Here, 'frustration' refers to mechanical incompatibility between neighboring grains that constrains easy slip."). Beyond that, no change is needed.

**Other Comments**

The discussion connecting the microstructural simulations to settlement parameterizations in snowpack models is one of the most valuable aspects of this paper. The authors also do a good job highlighting the role of missing microstructural descriptors such as bond size (Hagenmuller et al., 2014a), connectivity (Schleef et al., 2014b; Schöttner et al., 2025), and inter-crystalline surface area. It might be helpful to indicate briefly which of these factors could most readily be incorporated in future work.

**Recommendation**

This is an excellent paper that represents a meaningful step forward in understanding snow rheology and densification from a microstructural perspective. I recommend **minor revision** to address the small clarifications noted above. Once these are implemented, I would be happy to see the paper accepted for publication in *The Cryosphere*.

---

## Author Comment (AC1)

**Authors' Response to Reviews of**

**Revisiting snow settlement with microstructural knowledge.**

Louis Védrine, Pascal Hagenmuller
*The Cryosphere,*
* * *
**Comment:** *Reviewers' Comment*,     Response: Authors' Response

**Comment:** *Reviewer 1: November 10, 2025 To the Editors/Authors, I have reviewed the manuscript "Revisiting snow settlement with microstructural knowledge" submitted by Louis Védrine and Pascal Hagenmuller to The Cryosphere. Overall, I find this to be a very strong and well-crafted study that makes a meaningful contribution to our understanding of the viscoelastic and microstructural controls on snow settlement. The work combines microstructure-based crystal plasticity modeling with a thoughtful analysis of previous creep data, and the synthesis presented here brings valuable physical clarity to a problem that has long been described in more empirical terms. The manuscript fits well within the aims and scope of The Cryosphere. It meets the journal's criteria for originality, scientific rigor, and clarity, and it provides new insights of practical importance to snowpack and firn modeling communities.*

**1. General Evaluation**

**Comment:** *The study is carefully designed and well executed. The modeling approach is technically sound, the connection to experiments is well argued, and the presentation is clear. The results convincingly show why seemingly linear settlement laws appear to work under typical Alpine conditions, while also explaining where and why these relationships may fail. I have only a few points that I believe should be clarified before publication. These are primarily matters of explanation rather than scientific substance.*

**Response:** We want to thank the Reviewer 1 for his positive and motivating feedback, and constructive suggestions that helped us to improve the quality of our paper. In the following, we provide detailed point-by-point answers to the general comments from the reviewer.

**2. Specific Comments**

**Comment:** *1. On the 1% strain and the strain-rate minimum Around line 175, the authors use 1% strain to determine the steady (minimum) creep rate. This is a reasonable and widely accepted choice: for many materials, including polycrystalline ice, the minimum creep rate occurs at about 1% strain. Jacka (1984) reported the minimum at roughly 0.6%, and Treverrow et al. (2012) found steady conditions near 1%. I suggest the authors briefly note this and include one or two references to show that their cutoff corresponds to the transition from transient to steady-state creep.*

**Response:** Reply: Indeed, these results are consistent with what is commonly observed for polycrystalline ice. We have added experimental references for comparison as follows, "In our simulations, the permanent viscoplastic regime is reached after a typical strain of 0.5%. This is consistent with the steady (minimum) strain rate of polycrystalline ice, commonly measured around 1%(Mellor and Cole, 1982; Jacka, 1984; Treverrow et al., 2012). Consequently, we conduct simulations up to 1% strain to determine the flow stress $\sigma_Y$."

**Comment:** *2. On the stress exponent and grain boundary sliding. Lines 307–308 state that the observed stress exponent ($n \approx 2$) does not require invoking grain boundary sliding (GBS). This conclusion seems plausible given the microstructure-based modeling. However, previous studies—such as Alley (1987) and Goldsby & Kohlstedt (1997, 2001)—have associated $n \approx 2$ with GBS in fine-grained ice. It would be helpful to acknowledge that this similarity exists and to explain that the present work offers an alternative physical explanation (based on constrained basal glide in a porous polycrystal) for the same effective exponent. Doing so would better situate the results in the context of existing literature.*

**Response:** We have clarified this discussion as: "Furthermore, this suggests that the reduction of the stress exponent from dense ice to snow does not require introducing grain boundary sliding (GBS) as an additional deformation mechanism. GBS was originally proposed by Alley (1987) and Goldsby and Kohlstedt (1997) to explain observations of low stress sensitivity at low solid fractions and is associated with a stress exponent of $n \approx 2$ in fine-grained ice (Goldsby and Kohlstedt, 2001). Our work shows that the low crystalline frustration in snow provides an alternative physical explanation for the reduced stress exponent observed in snow."

**Comment:** *3. On crystal orientation and slip systems The model includes basal, prismatic, and pyramidal slip systems. Because basal slip dominates and is rotationally symmetric about the c-axis, the specific a-axis orientation typically has little effect on mechanical behavior at these stress levels. Still, it would be worth clarifying whether the simulations explicitly included full crystal orientations, or if c-axis randomization alone was used. A short note referencing Schulson & Duval (2009) or Weikusat et al. (2017) would be helpful to reassure readers that this simplification is appropriate.*

**Response:** Our simulations explicitly include full crystal orientations; both the c-axis and a-axes are randomly sampled from a uniform distribution. We have clarified this point as follows: "Then, each crystal was assigned a full crystallographic orientation (c-axis and a-axes) randomly sampled from an isotropic distribution."

**Comment:** *4. On the word "frustration" The term "frustration" appears several times and is used correctly in the mechanical sense—referring to geometric or kinematic incompatibility between neighboring grains that limits basal glide. For readers unfamiliar with materials-science terminology, I suggest adding a brief clarification at its first occurrence (e.g., "Here, 'frustration' refers to mechanical incompatibility between neighboring grains that constrains easy slip."). Beyond that, no change is needed.*

**Response:** Indeed, reviewer 2 required clarifications about this term from material-science terminology. We have clarified the use of the term "frustration" as follows: "Here, 'frustration' refers to the mechanical incompatibility between neighboring crystals that constrains soft deformation mechanisms."

**3. Other Comments**

**Comment:** *The discussion connecting the microstructural simulations to settlement parameterizations in snowpack models is one of the most valuable aspects of this paper. The authors also do a good job highlighting the role of missing microstructural descriptors such as bond size (Hagenmuller et al., 2014a), connectivity (Schleef et al., 2014b; Schöttner et al., 2025), and inter-crystalline surface area. It might be helpful to indicate briefly which of these factors could most readily be incorporated in future work.*

**Response:** As mentioned in lines 385-392, we chose not to introduce additional microstructural descriptors in order to maintain compatibility with existing experimental datasets and with detailed snowpack models, in which the microstructure is typically reduced to its ice fraction. From an experimental standpoint, integrating such descriptors into settlement parameterizations is relatively straightforward, for instance through the analysis of

topology. However, incorporating these descriptors into snow models is more challenging, as their temporal evolution must also be predicted. In the current version of Crocus (Vionnet et al., 2012; Lafaysse et al., 2017), the state variables related to the snow microstructure are density, specific surface area and sphericity. Unfortunately, these quantities do not contain information about the topology or the intercrystalline surface, and their use is not straightforward for physically based modelling. A first step could be the development of empirical relationships derived from in-situ measurements during settlement experiments. This point has been added as "Therefore, monitoring these indicators during in-situ compression tests (e.g. Bernard et al., 2023) could help identify their evolution laws." The integration of the inter-crystalline surface area appears more complex, as mentioned on lines 403-406 . Capturing both the geometry and the crystallographic structure of the microstructure requires coupling micro-CT tomography with DCT. Substantial work will be needed to understand the evolution of inter-crystalline surface area and to relate it to metamorphism.

**4. Recommendation**

**Comment:** *This is an excellent paper that represents a meaningful step forward in understanding snow rheology and densification from a microstructural perspective. I recommend minor revision to address the small clarifications noted above. Once these are implemented, I would be happy to see the paper accepted for publication in The Cryosphere.*

**References**

R. B. Alley. Firn densification by grain-boundary slidinf: a first model. *Le Journal de Physique Colloques*, 48(C1):C1–256, Mar. 1987.

A. Bernard, P. Hagenmuller, M. Montagnat, and G. Chambon. Disentangling creep and isothermal metamorphism during snow settlement with X-ray tomography. *Journal of Glaciology*, 69(276):899–910, Aug. 2023.

D. L. Goldsby and D. L. Kohlstedt. Grain boundary sliding in fine-grained Ice I. *Scripta Materialia*, 37(9): 1399–1406, Nov. 1997.

D. L. Goldsby and D. L. Kohlstedt. Superplastic deformation of ice: Experimental observations. *Journal of Geophysical Research: Solid Earth*, 106(B6):11017–11030, 2001.

T. H. Jacka. The time and strain required for development of minimum strain rates in ice. *Cold Regions Science and Technology*, 8(3):261–268, Mar. 1984.

M. Lafaysse, B. Cluzet, M. Dumont, Y. Lejeune, V. Vionnet, and S. Morin. A multiphysical ensemble system of numerical snow modelling. *The Cryosphere*, 11(3):1173–1198, May 2017.

M. Mellor and D. M. Cole. Deformation and failure of ice under constant stress or constant strain-rate. *Cold Regions Science and Technology*, 5(3):201–219, Mar. 1982.

A. Treverrow, W. F. Budd, T. H. Jacka, and R. C. Warner. The tertiary creep of polycrystalline ice: experimental evidence for stress-dependent levels of strain-rate enhancement. *Journal of Glaciology*, 58(208):301–314, Jan. 2012.

V. Vionnet, E. Brun, S. Morin, A. Boone, S. Faroux, P. Le Moigne, E. Martin, and J.-M. Willemet. The detailed snowpack scheme Crocus and its implementation in SURFEX v7.2. *Geoscientific Model Development*, 5 (3):773–791, May 2012.

---

## Author Comment (AC2)

**Authors' Response to Reviews of**

**Revisiting snow settlement with microstructural knowledge.**

Louis Védrine, Pascal Hagenmuller
*The Cryosphere,*
* * *
Comment: *Reviewers' Comment*,    Response: Authors' Response

**Comment:** *Reviewer 2: November 25, 2025 This is an interesting work dealing with the micromechanical modelling of the rheology of snow with various microstructures, mostly described by the solid volume fraction. The paper fits the scope of EGU sphere. There are a number of points that are worth to strengthen in the actual manuscript, therefore I recommend publication after major revisions (details below) have been taken into account.*

**Response:** We thank the reviewer 2 for his valuable comments and constructive suggestions that helped us to improve the quality of our paper. In the following, we provide detailed point-by-point answers to the comments from the reviewer.

**1. Comments**

**Comment:** *Equ (1) and (2): please indicate which strain-rate and stress components represents $\sigma$ and $\dot{\varepsilon}$. Line 32, and also later in the manuscript, I am not expert in snow rheology but using the term "Glen's law" for snow is misleading. Glen's flow law is used for ice, which is considered incompressible and therefore it is the deviatoric stress that enter into the formula. Here, snow deformation is not isochoric, so it departs from Glen's flow law used in glaciology.*

**Response:** Equations (1) and (2) correspond to the generic 1D formulation of uniaxial compression under free-lateral-boundary conditions. In our case, we choose the loading direction to be the $zz$ direction. $\sigma$ and $\dot{\varepsilon}$ refers to the vertical (zz-direction) stress and strain rate. The power law is sometimes, albeit improperly, referred to as a "Glen's-type law" in snow mechanics (Scapozza and Bartelt, 2003; Schleef et al., 2014; Sundu et al., 2024). We fully agree that for ice, which is incompressible, only the deviatoric stress contributes to the flow. To avoid any ambiguity, we have removed the term "Glen's law" from the manuscript.

**Comment:** *Line 44-45 "Rheological models, for instance, the Maxwell, Kelvin, or Burger models": the link with previous sentence is not clear. I think the authors should write that equ 1 and 2 are for steady state creep, whereas the Maxwell, Kelvin or Burgers models account for the transient creep regime.*

**Response:** Indeed, Maxwell, Kelvin, and Burger-type models are commonly used to represent the transient creep regime. In the context of snow compaction, the models cited here include a viscosity term that represents the steady-state viscous flow, and aim to reproduce the long-term evolution of snow density in creep experiments. For example, in (Gorynina et al., 2024; Huo et al., 2024; Bahaloo et al., 2024), Maxwell and Burger models are used to represent both the transient regime and the permanent viscous flow. We have clarified that Equations 1 and 2 correspond to steady-state creep. We also specify that the rheological models cited here are used to describe the viscous regime. "Rheological models, for instance the Maxwell, Kelvin, or Burger models, are also used to reproduce the permanent viscous flow."

**Comment:** *Line 115 The authors should clearly indicate the bibliographic references for the tomographic images*

*used in fig 1, either in the figure or in its legend. Same for fig 5*

**Response:** The references for the images used were previously indicated only in the supplementary material; they are now provided in a Table.

**Comment:** *Line 124 explain what SSA means.*

**Response:** The SSA [m²/kg] is the specific surface area, defined as the surface area per unit snow mass. Density and specific surface area (SSA) are classical indicators used to characterize the microstructure. The acronym and its definition have been added in the revised manuscript. "These binary ice-air images encompass the majority of snow types encountered in seasonal snowpacks and span a wide range of density and specific surface area (SSA), i.e., the surface area per unit snow mass."

**Comment:** *Line 131 "Air was modelled as an infinitely soft elastic medium": having infinite mechanical contrasts in the spectral (FFT) scheme often prevents the model to converge. Please detail.*

**Response:** Indeed, the number of iterations required to achieve convergence with the original Moulinec and Suquet (1992) FFT-based algorithm is inversely proportional to the contrast between the phases. This limitation has strongly constrained the study of viscoplastic behavior, particularly at low solid-fractions. In this study, the solver (AMITEX_FFTP) uses a modified Green operator based on a hexahedral Finite Difference scheme (Willot, 2015), which is equivalent to using linear hexahedral finite elements with reduced integration (Schneider et al., 2017), combined with Anderson convergence acceleration (Anderson, 1965). This approach enables the analysis of highly porous materials (infinite contrast with the solid matrix) down to solid fractions of 0.1. We have now clarified the specific features of the solver in the manuscript, which allow the study of low solid fractions with high phase contrast. "This solver employs a modified Green operator based on a hexahedral finite difference scheme (Willot, 2015), equivalent to using linear hexahedral finite elements with reduced integration (Schneider et al., 2017), and an Anderson convergence acceleration technique (Anderson, 1965), enabling the analysis of porosities (infinite contrast with the solid matrix) up to 90%."

**Comment:** *Lines 150-151 for a vertical compression, $\varepsilon_{zz}$ should be negative, not positive*

**Response:** As in soil mechanics, where the problem is purely compressive, both the strain rate and stress are defined as positive. This convention has now been clarified.

**Comment:** *Section 2.1.2 the rheology used at the crystal scale should be clarified. I guess it is of Maxwell type, i.e. without considering transient creep response at the crystal scale as in Suquet et al 2012?*

**Response:** The rheology of the crystal plasticity model is provided in Appendix A. The viscoplastic deformation of each slip system is modeled using a power law. Indeed, unlike Suquet, who considers the transient creep response at the crystal scale, we focus on the steady-state response. We therefore assume that the critical stresses $\tau_0^{(k)}$ are constant in time (no hardening) and equal to the steady-state values reported by Suquet. This has been clarified in as, " We therefore used the stationary parametrization of Suquet et al. (2012), calibrated to reproduce the viscoplastic behaviour of dense polycrystalline ice ($\sigma_0(\Phi = 1) = 272$ MPa and $n_{suquet} = 2.94$) as proposed by Budd and Jacka (1989) and Castelnau et al. (1996) at $T_0 = 263.15$ K.". And in appendix, ""As we only focus on the steady-state response and to simplify comparison, the critical stresses $\tau_0^{(k)}$ are assumed to be constant with time (no hardening), and the reference shear rate is set to $\dot{\gamma}_0^{(k)} = 1$ s$^{-1}$. We calibrate the values of the Critical Resolved Shear Stresses (CRSS) to obtain the viscoplastic law of dense polycrystalline ice proposed by Budd and Jacka (1989); Castelnau et al. (1996) at -10°C (A=7.8 $\times 10^{-8}$ MPa$^{-1}$ s$^{-3}$ and $n_{poly} = 2.94$ the parameters of the Glen law for polycrystalline ice $\dot{\epsilon} = A\sigma^{n_{poly}}$)."

**Comment:** *The microstructures indicated in fig 2 and 3 show very few contacts between the crystals. This should*

*quite heavily influence the overall rheology. It would be interesting that the authors add a comment on this, and provide a statistical quantification of the contact area, compared for example with the Voronoi microstructures used in the Vedrine (Acta Mater 2025) paper.*

**Response:** The low contact surface between crystals is at the origin of their low frustration. The contiguity parameter $\beta$, defined as the ratio of inter-crystalline surfaces to the total surface (inter-crystalline surfaces + free surfaces) (Underwood, 1970), was computed for all the microstructures and is reported in Appendix B. For the snow microstructures considered,$\beta \approx 0.1$, meaning that, on average, only 10% of a crystal's perimeter is in contact with another crystal. For Voronoi-type microstructures, where the pore size is comparable to the crystal size,$\beta$ ranges from 0.2 to 0.45 for solid fractions between 0.3 and 0.6 This indicates that Voronoi microstructures have a larger inter-crystalline surface and are more frustrated. The low frustration of the snow microstructures studied helps explain their low stress exponent.The reference to Appendix B has been added in the revised manuscript, and a comparison with Voronoi-type microstructures has also been included in Appendix B. "Note that the surface ratio obtained for these snow microstructures is significantly lower than that for Voronoi-type microstructures with pores and crystals of similar size. Indeed, for a solid fraction of $\phi = 0.4$, the snow microstructures exhibit a surface ratio of approximately $\beta \approx 0.1$, whereas the Voronoi microstructures exhibit a value of about $\beta \approx 0.35$."

**Comment:** *Line 158, what do you mean with "The numerical integration of crystal plasticity is computationally expensive"? Provide details?*

**Response:** Because the constitutive law is nonlinear, a Newton–Raphson algorithm is used to solve it at each time step. As the problem involves a large number of degrees of freedom (12 coupled linear systems), the computational cost is significant: approximately 12 hours on sample presented in Fig. 3 of size 250x250x250 voxels, on a high-performance computer with 256 Processeur Intel® Xeon® Gold 6130 cores. We have included these details in the revised manuscript as, "The numerical integration of crystal plasticity is computationally expensive (approximately 12 hours on sample presented in Fig. 3 of size 250x250x250 voxels, on a high-performance computer with 256 Processeur Intel® Xeon® Gold 6130 cores), but the solver benefits from the MPI implementation."

**Comment:** *Line 171 I don't understand why you speak about "an elastic regime". Here there is no threshold for the activation of plasticity, so viscoplastic deformation should occur whatever the prescribed stress level.*

**Response:** Although the yield stress is indeed zero in our formulation, the plastic contribution remains negligible at low stress levels and short times, so that the response is effectively elastic. We have clarified this point in the revised manuscript. "First, the axial stress $\sigma_{zz}$ increases linearly with strain in an almost elastic regime. Indeed, at very low stress levels and short times, the viscoplastic strain is negligible."

**Comment:** *Along the same line, I think you should not speak (line 172 but also elsewhere in the manuscript) about a "yield stress", which refer to some stress threshold to activate plasticity. A better word is probably "flow stress"?*

**Response:** Indeed, we use the term "yield stress" to refer to the steady-state stress. To avoid any ambiguity, the term "flow stress" is now used in the revised manuscript.

**Comment:** *I find lines 201-204 unnecessary (delete?)*

**Response:** This paragraph explains in plain text the strategy used to recover the couple of parameters ($\sigma_0$, n) from experimental points ($\sigma$,$\dot{\epsilon}$). We believe it is necessary for the wide audience of The Cryosphere.

**Comment:** *As written, equation (6) is not a "minimization problem"*

**Response:** You are right. The minimisation problem associated with the error $\mathcal{L}$ defined in Equation 6 is now explicitly written.

**Comment:** *Lines 232-234 are not clear. What do you mean with "BC have a negligeable effects"? Effects on what? BC effects is generally considered with care specially because they can significantly affect model results.*

**Response:** By boundary conditions, we specifically refer to the lateral boundary conditions. In our simulations, we conduct uniaxial compression (zero average lateral stresses, $\sigma_{xx} = \sigma_{yy} = 0$) to obtain the 1D constitutive law, whereas natural conditions are closer to an oedometer test (with lateral confinement). However, Wautier et al. (2017) demonstrated that, due to the high compressibility of snow and the predominance of the hydrostatic component of the stress (resulting from a low effective Poisson ratio), the lateral boundary conditions have only a negligible influence. We have clarified this as: "Lateral boundary conditions have a negligible effect (Wautier et al., 2017) because of the predominance of the hydrostatic stress state. Consequently, free and confined boundary conditions are not generally distinguished in these models."

**Comment:** *The word "frustration" is often used in the manuscript, but it is not common in the field of micromechanics. Please detail what is meant with it.*

**Response:** Indeed, reviewer 1 also required clarifications about this term from material-science terminology. We will use the clear definition proposed by reviewer : "Here, 'frustration' refers to the mechanical incompatibility between neighbouring crystals that constrains soft deformation mechanisms."

**Comment:** *Line 245 it could be interesting to compare your results with the lower bound (uniform stress within the specimen) as it also leads to n=2 too. How far is the stress heterogeneous (ex. Standard deviation) wrt to volume fraction? & 256-260 I have no idea from where comes the statement "The solid fraction sensitivity m relates to the heterogeneity of viscoplastic deformations within the sample" which sounds weird to me. Please explain or provide the bibliographic reference. And next sentence, the "m" value computed for linear behaviour (such as thermo-elasticity) is probably very different from the "m" computed in the viscoplastic regime. Could you compare both?*

**Response:** Indeed, by assuming a uniform stress within the matrix, one obtains $n = 2$. This assumption also implies that $\sigma_0 = \sigma_{0,\text{ice}}\, \phi^{m=1}$, which significantly differs from our findings on numerical homogenization (m=3.96) creep experiments $m = 3.195$. All in all, our results suggest that the stress field deviates significantly from a uniform stress field. To support this statement, we have added Figure 1 in the revised manuscript, showing the distribution of the von Mises equivalent stress for a snow microstructure $\Phi$=0.47). A wide dispersion of stresses is observed within the solid matrix, with a coefficient of variation of 0.867(Fig.1). In comparison, for Voronoi-type microstructures ( Vedrine et al. paper [Acta Mater, 2025]), Fig. 1), the coefficient of variation is 0.709. The snow microstructure is characterised by significantly higher extreme values, with about 1.7% of the voxels exceeding three times the mean equivalent stress, compared with only 0.7% for the Voronoi microstructures. These results are consistent with the interpretation proposed by Roberts and Garboczi (2002); Bruno et al. (2011), that $m$ can be viewed as an indicator of microstructural heterogeneity linked to morphology. As mentioned in the submitted manuscript, measured values of $m$ in snow appear to depend on the studied process. Nevertheless, the underlying cause remains unclear, and differences reported by different groups may partly arise from experimental protocols or analysis methods.

**Comment:** *Line 270, "suggesting that under loading, the microstructure tends to optimize itself"?? You mean that there is some microstructure evolution during deformation, but this is not considered in your computation, so what might be the effect? And I don't see the link with the over- or under-estimation of the reference stress indicated in the previous sentence.*

**Response:** The evolution of the microstructure is indeed not accounted for in our simulations. However, we have access

[Figure]

Figure 1: Distribution of the normalised local Von Mises equivalent stress in a voronoi type and snow microstructure (Φ=0.5).

to a series of microstructural images obtained during an in-situ compression experiment, which allows us to evaluate, using our numerical simulations, the evolution of the instantaneous material properties over time. We have now clarified that the term model refers to the fitted macroscopic law for s0 and not to the numerical simulations themselves. This distinction has also been explicitly stated as "Similarly, the reference stress simulated for DF series studied by Bernard (2023) cannot be perfectly fitted with a power law with a constant exponent . The values of m tends to decrease with the densification."

**Comment:** *Section 3.2 Why do you call this section "experimental data-driven model"?? I don't understand what it means. And why "data-driven"?*

**Response:** Indeed, we do not use machine learning, neural networks, or parameter identification on a large dataset, and the term "data-driven" could therefore be misleading. We refer to Section 3.2 as the "experimental data-driven model" because we exploit the diversity of experimental conditions (e.g., type of test: imposed stress or strain-rate, strain rate, solid fraction, temperature) to analyze slow snow compaction tests and identify the stress exponent $n$ that accounts for the unexplained dispersion between these data. In this sense, a single experiment would not provide this information; it is the global analysis of the dataset that enables its extraction. To remove any confusion, we have renamed Section 3.2. "Model calibrated on mechanical tests".

**Comment:** *Fig 8 and lines 338-354. First, I don't understand the figure, as you represent in colour the R values (differences between V25 and Br92 strain-rates) but the points and dash lines are stress vs volume fraction. Why do you say then that" This agreement supports the validity of the V25 model" (line 349) as you don't show that V25 reproduces the Col de Porte data but only that it is in agreement with Br92 (without showing*

**that Br92 reproduce Col de Porte data)?**

**Response:** Figure 8 shows the $R$ values (differences between the V25 and Br92 strain rates) for different stress–solid fraction pairs. We have also added the measured stress versus solid fraction points obtained from stratigraphic observations at three different sites, in order to illustrate the typical loading conditions associated with a given solid fraction at the site level. The original compaction law proposed by Navarre (1975), based on observations at Col de Porte, was recalibrated by Brun (1992) at the same site. Brun (1992) showed that incorporating Navarre's compaction settlement law provides a good representation of snow height evolution, and concluded that the Crocus model is very efficient in accurately simulating the temporal evolution of snow-cover stratigraphy throughout the winter season. Vionnet et al. (2012) also compared the global properties (snow depth and snow water equivalent, SWE) simulated by Crocus to measurements at Col de Porte and obtained satisfactory scores. Based on these results, we can assume that the Crocus compaction law predicts, on average, appropriate strain rates for the Col de Porte site. Consequently, we have nuanced our statement as follows: "Notably, the two models yield equivalent results for the solid fraction–stress couples most commonly observed at the Col de Porte, where the Br92 model has been extensively calibrated (Navarre, 1975; Brun et al., 1992; Vionnet et al., 2012; Lafaysse et al., 2017). This consistency indicates that V25 behaves similarly to Br92 under typical field conditions, and therefore suggests that the V25 model is also likely to reproduce the snow compaction observed at Col de Porte."

**Comment:** *Same for the last sentence "Br92 over-estimates the strain rate for $\Phi > 0.08$ and stresses smaller than those observed at col de Porte, Br92 under-estimates the strain rate for very low-density samples ($\Phi < 0.08$) and for stresses higher than the "standard" ones" as you don't show direct comparison between modelled stress and field stress.*

**Response:** In a snow model, the stress acting on a given layer is prescribed by the problem: it is directly related to the gravitational constant multiplied by the vertical integral of the snow water equivalent (SWE) [m²/kg] overlying that layer. For a given stress and solid fraction, it is therefore possible to directly compare the deformation rates predicted by the two parameterisations considered. We clarified this as follows: "The overburden stress is defined as the gravitational constant multiplied by the vertical integral of the snow water equivalent (SWE) overlying the layer." and "BR92 predicts higher strain rates than V25 for densities above $\Phi > 0.08$ and for stresses lower than those observed at Col de Porte. Conversely, for very low density samples ($\Phi < 0.08$) and for stresses higher than the "standard" ones Br92 predicts lower strain-rate compared to V25."

**Comment:** *Fig 9, 10, 11, you write line 368 "This result aligns with observations" but the observations are not shown. . .*

**Response:** As far as we know, there are no systematic field observations in the literature that isolate snow compaction under controlled initial and loading conditions (temperature and overburden stress), particularly in situations such as heavy snowfalls, wind-packed surface layers, or deep firn. It is therefore not possible to directly compare our new parameterisation to a field observation. Instead, we compare our formulation to the BR92 parameterisation, which has been repeatedly tested against extensive observational datasets and is widely used in operational snow models (Brun et al., 1992; Vionnet et al., 2012; Lafaysse et al., 2017) As clarified in the revised manuscript, Figs. 9, 10, and 11 illustrate this comparison in three configurations where existing compaction laws have been questioned in previous studies:
**1. Fresh-snow settlement.** Several studies (Lundy et al., 2001; Steinkogler et al., 2009; Wever et al., 2015) et al., 2001; showed that SNOWPACK underestimates the compaction of recent snow. Quéno et al. (2016) reached similar conclusions with AROME–Crocus when comparing simulated and observed snow depth. These studies point to an insufficient densification rate after snowfall.
**2. Basal Arctic snow layers:** Woolley et al. (2024) reported that models tend to underestimate the slow

densification of basal Arctic layers, further questioning existing compaction parameterisations.

**3. Deep firn:** Touzeau et al. (2018) and Verjans et al. (2019) reported an underestimation of the densification rate of dense snow and firn in glacier accumulation zones. To account for this issue, van Kampenhout et al. (2017) decreased the viscosity in the BR92 model. We have clarified these points and add references to support the observation mentioned in the revised manuscript.

**Comment:** *Line 404, the "crystalline structure" of ice Ih is very well known (space group P 63/mmc, etc).*

**Response:** We have clarified in line 404 that we refer to the crystalline structure (i.e., crystal orientation and inter-crystalline surface) of depth hoar, and not to the properties of ice at the crystal scale. "Due to the lack of knowledge about the exact crystalline structure (i.e., crystal orientation and intercrystalline surface) of depth hoar, we cannot currently test this intuition."

**Comment:** *Lines 416-421 I don't understand why you state that "our model is one-dimensional" as your FFT computations and microstructures are 3d, so you deal with full 3d problems...*

**Response:** The objective of this work is to unify existing snow settlement models with laboratory and numerical experiments based on microstructures. Indeed, we use 3D simulations and microstructures, but we propose only a 1D law to represent uniaxial snow compaction. This choice facilitates comparison with existing models and ensures that the law can be easily implemented in current snow models. An extension of the constitutive law to the 3D case could enable applications to compaction under shear or interactions with structures. Extending the model to 3D would require accounting for the effect of stress triaxiality on the constitutive law, and is therefore left for future studies.

**Comment:** *Line 441, "In fact, the linear viscosity $\eta$, a function of solid fraction (which itself is a function of stress), can artificially introduce non-linearity"? Do you really mean that the volume fraction depends on the stress? Perhaps you mean cumulative strain?*

**Response:** Yes, we really mean that, on certain sites, solid fraction is correlated to stress (see Fig. 8). This is related to the storyline of the snowpack life. For instance, in the Arctic, where snowfall is low and wind is strong, for identical solid fractions, the snowpack is less loaded than in the Alpine snowpacks. Thus, each site exhibits a characteristic stress–density relationship. However, as discussed in Section 3.2, assuming a linear law does not capture laboratory compaction experiments. However, in snow models, the viscosity is a fitted parameter that depends on the solid fraction, which implicitly includes both the material's viscosity variation with $\phi$ and the typical evolution of stress with density. Assuming that $\sigma$ follows a function $s(\phi)$, we have shown that the constitutive law for snow can be expressed as

$$\dot{\varepsilon} = \left( \frac{\sigma}{\sigma_0} \right)^{2.15}, \tag{1}$$

whereas in typical snow models, the same behavior is written as

$$\dot{\varepsilon} = \frac{\sigma}{\eta}, \qquad \eta = \frac{\sigma_0^{2.15}}{s(\phi)^{1.15}}. \tag{2}$$

This demonstrates that the fitted viscosity in snow models effectively encodes the non-linear dependence of stress on density through $s(\phi)$. We have clarified this sentence as: "In fact, the empirical linear viscosity $\eta$ in snowpack models implicitly accounts for both the material's viscosity variation with $\phi$ and the typical evolution of stress with density, which are not included in the linear stress term of the model. As a result, it can artificially introduce non-linearity into the constitutive equation."

**Comment:** *Line 455 the indicated elastic behaviour is not isotropic but transverse isotropic*

**Response:** Corrected.

**Comment:** *A good point is that the Vedrine et al. paper [Acta Mater, 2025] that is often cited in this manuscript has been accepted for publication.*

**References**

D. G. Anderson. Iterative Procedures for Nonlinear Integral Equations. *Journal of the ACM*, 12(4):547–560, Oct. 1965.

H. Bahaloo, F. Forsberg, H. Lycksam, J. Casselgren, and M. Sjödahl. Material mapping strategy to identify the density-dependent properties of dry natural snow. *Applied Physics A*, 130(2):141, Jan. 2024.

A. Bernard. *Etude multiéchelle de la transition ductile-fragile dans la neige*. phdthesis, Université Grenoble Alpes [2020-....], Mar. 2023.

E. Brun, P. David, M. Sudul, and G. Brunot. A numerical model to simulate snow-cover stratigraphy for operational avalanche forecasting. *Journal of Glaciology*, 38(128):13–22, Jan. 1992.

G. Bruno, A. M. Efremov, A. N. Levandovskyi, and B. Clausen. Connecting the macro- and microstrain responses in technical porous ceramics: modeling and experimental validations. *Journal of Materials Science*, 46(1):161–173, Jan. 2011.

W. F. Budd and T. H. Jacka. A review of ice rheology for ice sheet modelling. *Cold Regions Science and Technology*, 16(2):107–144, July 1989.

O. Castelnau, P. Duval, R. A. Lebensohn, and G. R. Canova. Viscoplastic modeling of texture development in polycrystalline ice with a self-consistent approach: Comparison with bound estimates. *Journal of Geophysical Research: Solid Earth*, 101(B6):13851–13868, 1996.

O. Gorynina, P. Bartelt, and G. Gorynin. One-Dimensional Visco-Elastic Finite Element Modeling of the Snow Creep, Apr. 2024.

H. Huo, Q. Chen, E. Xiao, H. Li, H. Xu, T. Li, and X. Tang. Long-Term One-Dimensional Compression Tests and Fractional Creep Model of Compacted Snow. *Cold Regions Science and Technology*, 228:104326, Dec. 2024.

M. Lafaysse, B. Cluzet, M. Dumont, Y. Lejeune, V. Vionnet, and S. Morin. A multiphysical ensemble system of numerical snow modelling. *The Cryosphere*, 11(3):1173–1198, May 2017.

C. C. Lundy, R. L. Brown, E. E. Adams, K. W. Birkeland, and M. Lehning. A statistical validation of the snowpack model in a Montana climate. *Cold Regions Science and Technology*, 33(2):237–246, Dec. 2001.

J. P. Navarre. *Modele unidimensionnel d'evolution de la neige deposee*. Société météorologique de France, 1975.

L. Quéno, V. Vionnet, I. Dombrowski-Etchevers, M. Lafaysse, M. Dumont, and F. Karbou. Snowpack modelling in the Pyrenees driven by kilometric-resolution meteorological forecasts. *The Cryosphere*, 10 (4):1571–1589, July 2016.

A. P. Roberts and E. J. Garboczi. Computation of the linear elastic properties of random porous materials with a wide variety of microstructure. *Proceedings of the Royal Society of London. Series A: Mathematical, Physical and Engineering Sciences*, 458(2021):1033–1054, May 2002.

C. Scapozza and P. A. Bartelt. The influence of temperature on the small-strain viscous deformation mechanics of snow: a comparison with polycrystalline ice. *Annals of Glaciology*, 37:90–96, Jan. 2003.

S. Schleef, H. Löwe, and M. Schneebeli. Hot-pressure sintering of low-density snow analyzed by X-ray microtomography and in situ microcompression. *Acta Materialia*, 71:185–194, June 2014.

M. Schneider, D. Merkert, and M. Kabel. FFT-based homogenization for microstructures discretized by linear hexahedral elements. *International Journal for Numerical Methods in Engineering*, 109(10):1461–1489, 2017.

W. Steinkogler, C. Fierz, M. Lehning, and F. Obleitner. Systematic Assessment of New Snow Settlement in Snowpack. *International Snow Science Workshop, Davos 2009, Proceedings*, pages 132–135, 2009.

K. Sundu, R. Ottersberg, M. Jaggi, and H. Löwe. A grain-size driven transition in the deformation mechanism in slow snow compression. *Acta Materialia*, 262:119359, Jan. 2024.

P. Suquet, H. Moulinec, O. Castelnau, M. Montagnat, N. Lahellec, F. Grennerat, P. Duval, and R. Brenner. Multi-scale modeling of the mechanical behavior of polycrystalline ice under transient creep. *Procedia IUTAM*, 3:76–90, 2012.

A. Touzeau, A. Landais, S. Morin, L. Arnaud, and G. Picard. Numerical experiments on vapor diffusion in polar snow and firn and its impact on isotopes using the multi-layer energy balance model Crocus in SURFEX v8.0. *Geoscientific Model Development*, 11(6):2393–2418, June 2018.

E. E. Underwood. *Quantitative Stereology*. Addison-Wesley, Reading, Mass.,, 1970.

L. van Kampenhout, J. T. M. Lenaerts, W. H. Lipscomb, W. J. Sacks, D. M. Lawrence, A. G. Slater, and M. R. van den Broeke. Improving the Representation of Polar Snow and Firn in the Community Earth System Model. *Journal of Advances in Modeling Earth Systems*, 9(7):2583–2600, 2017.

V. Verjans, A. A. Leeson, C. M. Stevens, M. MacFerrin, B. Noël, and M. R. van den Broeke. Development of physically based liquid water schemes for Greenland firn-densification models. *The Cryosphere*, 13(7): 1819–1842, July 2019.

V. Vionnet, E. Brun, S. Morin, A. Boone, S. Faroux, P. Le Moigne, E. Martin, and J.-M. Willemet. The detailed snowpack scheme Crocus and its implementation in SURFEX v7.2. *Geoscientific Model Development*, 5 (3):773–791, May 2012.

A. Wautier, C. Geindreau, and F. Flin. Numerical homogenization of the viscoplastic behavior of snow based on X-ray tomography images. *The Cryosphere*, 11(3):1465–1485, June 2017.

N. Wever, L. Schmid, A. Heilig, O. Eisen, C. Fierz, and M. Lehning. Verification of the multi-layer SNOWPACK model with different water transport schemes. *The Cryosphere*, 9(6):2271–2293, Dec. 2015.

F. Willot. Fourier-based schemes for computing the mechanical response of composites with accurate local fields. *Comptes Rendus Mécanique*, 343(3):232–245, Mar. 2015.

G. J. Woolley, N. Rutter, L. Wake, V. Vionnet, C. Derksen, R. Essery, P. Marsh, R. Tutton, B. Walker, M. Lafaysse, and D. Pritchard. Multi-physics ensemble modelling of Arctic tundra snowpack properties. *The Cryosphere*, 18(12):5685–5711, Dec. 2024.